# Suggesting Variable Order for Cylindrical Algebraic Decomposition via Reinforcement Learning

**Fuqi Jia**[1, 3][*], **Yuhang Dong**[2, 3][*], **Minghao Liu**[1, 3], **Pei Huang**[4], **Feifei Ma**[2, 3][†], and **Jian Zhang**[1, 3][†]

[1]State Key Laboratory of Computer Science, Institute of Software, Chinese Academy of Sciences, Beijing, China
[2]Laboratory of Parallel Software and Computational Science, Institute of Software, Chinese Academy of Sciences, Beijing, China
[3]University of Chinese Academy of Sciences, Beijing, China
[4]Stanford University, Stanford, USA
{jiafq,liumh,maff,zj}@ios.ac.cn, dongyuhang22@mails.ucas.ac.cn,
huangpei@stanford.edu

## Abstract

Cylindrical Algebraic Decomposition (CAD) is one of the pillar algorithms of symbolic computation, and its worst-case complexity is double exponential to the number of variables. Researchers found that variable order dramatically affects efficiency and proposed various heuristics. The existing learning-based methods are all supervised learning methods that cannot cope with diverse polynomial sets. This paper proposes two Reinforcement Learning (RL) approaches combined with Graph Neural Networks (GNN) for Suggesting Variable Order (SVO). One is GRL-SVO(UP), a branching heuristic integrated with CAD. The other is GRL-SVO(NUP), a fast heuristic providing a total order directly. We generate a random dataset and collect a real-world dataset from SMT-LIB. The experiments show that our approaches outperform state-of-the-art learning-based heuristics and are competitive with the best expert-based heuristics. Interestingly, our models show a strong generalization ability, working well on various datasets even if they are only trained on a 3-var random dataset. The source code and data are available at https://github.com/dongyuhang22/GRL-SVO.

## 1 Introduction

As learned in school, we know how to answer the question of whether a quadratic equation for $x$ has a real root. For example,

$$x^2 + bx + c = 0,$$

where $b, c$ are unknowns. We can answer it by checking whether the discriminant is non-negative, i.e., $b^2 - 4c \geq 0$. What if the number and degree of variables increase, and the formula involves the combination of the universal quantifier ($\forall$), existential quantifier ($\exists$), and logical operators (and($\wedge$), or($\vee$), not($\neg$))? Checking whether polynomials satisfy some mathematical constraints is a difficult problem and has puzzled mathematicians since ancient times. Until the 1930s, Alfred Tarski [1] answered the question by proving that the theory of real closed fields admits the elimination of quantifiers and gives a quantifier elimination procedure. Unfortunately, the procedure was impractical due to its non-elementary complexity. In 1975, George Collins discovered the first relatively efficient algorithm, Cylindrical Algebraic Decomposition (CAD) [2]. Currently, CAD (and variants) has

---

[*]These authors contributed equally.
[†]Corresponding authors.

37th Conference on Neural Information Processing Systems (NeurIPS 2023).

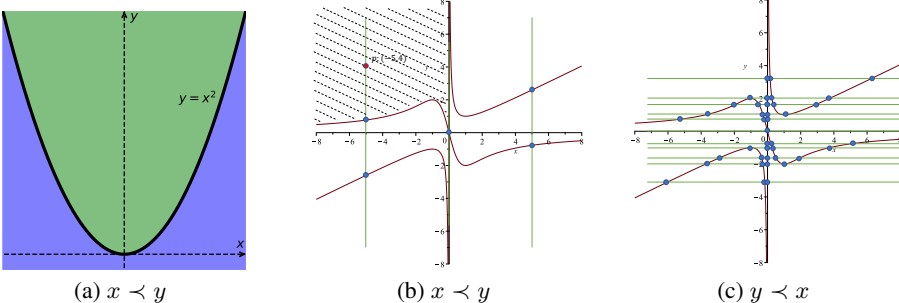

| (a) $x \prec y$ | (b) $x \prec y$ | (c) $y \prec x$ |

Figure 1: Examples of CAD. Figure 1a shows cells of $\{y - x^2\}$. Figure 1b and 1c are CAD with different variable orders on $\{x^3 y + 4x^2 + xy, -x^2 + 2xy - 1\}$.

become one of the most fundamental algorithms in symbolic computation and is widely used in computational geometry [3], robot motion planning [4], constraint programming [5, 6, 7].

More precisely, CAD is an algorithm that eliminates (technically called *project*) variables one by one and finally results in a list of regions that are sign-invariant to the polynomials (technically called *cells*). CAD provides an efficient quantifier elimination in real space, thereby enabling the solution of various problems related to polynomials. For example, the space of the discriminant of the quadratic equation (i.e., $b^2 - 4c \geq 0$) can be the combination of satisfied cells. We provide a detailed description in Appendix A. Due to its powerful analytical ability and great versatility, it is also accompanied by huge limitations. The theoretical worst complexity is double exponential to the number of variables [2].

Researchers have conducted in-depth studies on improving its efficiency. According to theoretical and practical research, there lives a very important conclusion that "variable order in CAD can be crucially important" [8, 9, 10, 11, 12, 13, 14, 15, 16]. The selection of variable orders has a great effect on the time, memory usage as well as the number of cells in CAD. As an example, [8] introduces a category of problems where one variable order leads to a result with double exponential complexity to the number of variables, while another order yields a constant-sized result.

In this paper, we present a Graph-based Reinforcement Learning for Suggesting Variable Order (GRL-SVO) approach for CAD. It has two variants: GRL-SVO(UP) (i.e., utilizing *project*) and GRL-SVO(NUP) (i.e., not utilizing *project*). GRL-SVO(UP) is integrated into the CAD and can select the "best" next projecting variable. Considering the high cost of interacting with the symbolic computation tool, we also propose a fast approach GRL-SVO(NUP), which will simulate the state transition (i.e., *project*) via two rules (update rule and delete rule). It can report a total variable order before the CAD process. To evaluate the effectiveness of the models, we conduct a dataset of random polynomial sets with 3 to 9 variables and collected instances from SMT-LIB [13, 17] to form a real-world dataset. Experimental results show that our approaches outperform state-of-the-art learning-based heuristics and are competitive with the best expert-based heuristics. GRL-SVO also exhibits a strong generalization capability. The models are only trained on a 3-var random dataset, but they still work well on other datasets.

## 2 Background and Related Work

In this section, we briefly introduce some basic definitions of CAD [2, 18]. We classify the previous works of SVO and give an overview of the techniques used.

### 2.1 Cylindrical Algebraic Decomposition (CAD)

A Cylindrical Algebraic Decomposition (CAD) is a decomposition algorithm of a set of polynomials in ordered $\mathbb{R}^n$ space resulting in finite sign-invariant regions, named *cells*. As shown in Figure 1a, there are 3 cells with different colors (two infinite regions and the curve), and any points in the cell lead to the same sign of $y - x^2$. Let $\mathbb{R}[\boldsymbol{x}]$ be the ring of polynomials in the variable vector $\boldsymbol{x} = [x_1, \cdots, x_n]$ with coefficients in $\mathbb{R}$ [9].

**Definition 1** (Cell). *For any finite set $Q \subseteq \mathbb{R}[x]$, a cell of $Q$ is defined as a maximally connected set in $\mathbb{R}^n$ where the sign of every polynomial in $Q$ is constant.*

CAD accepts a set of polynomials and a fixed variable order and mainly consists of three running phases: *project*, *root isolate*, and *lift*. The *project* phase eliminates a variable of a polynomial set once at a time. It will result in a new *projected* polynomial set that carries enough information to ensure possible decomposition. After repeating calls to *project*, CAD constitutes a step-like (from n-1 variables to 1 variable) set of *projected* polynomials. The *root isolate* procedure isolates all roots of the univariate polynomial set, and the roots split $\mathbb{R}$ into some segmentations. The *lift* phase samples in the segmentations and assigns the sampled value to the former *projected* polynomial set so that the former polynomial set will become a univariate polynomial set. After repeating *root isolate* and *lift* $n - 1$ times, CAD reconstructs the entire space via a set of cells characterized by the sample points. Since the CAD process and the *project* operators are not prerequisites to understand our approach, we arrange more details in Appendix A. Here, we exemplify the process and the effect of different variable orders.

**Example 2.1.** *Consider a polynomial set $\{x^3y + 4x^2 + xy, -x^2 + 2xy - 1\}$ as in Figure 1b and 1c.*

*CAD process.* *Assume that the variable order is $x \prec y$. CAD eliminates $y$ first and results in a polynomial set $\{x, x^2 + 1, x^4 + 10x^2 + 1\}$ (i.e., project phase). The polynomial set has only one root $x = 0$ (i.e., root isolation phase). Then $\mathbb{R}$ will be split into three segmentations: $\{x < 0, x = 0, x > 0\}$. We sample $\{x = -5, x = 0, x = 5\}$ and result in three different polynomial sets: $\{-130y + 100, -10y - 26\}$, $\{-1\}$, and $\{130y + 100, 10y - 26\}$ (i.e., lift phase). Let's take the first polynomial set as an example, and it has two roots, i.e., $\{\frac{10}{13}, -\frac{13}{5}\}$ (i.e., root isolation phase). Then $\mathbb{R}$ will be split into five segmentations: $\{y < -\frac{13}{5}, y = -\frac{13}{5}, -\frac{13}{5} < y < \frac{10}{13}, y = \frac{10}{13}, y > \frac{10}{13}\}$. As shown in Figure 1b, the sample red point $(-5, 4)$ can represent a sign-invariant region, the whole shaded area (i.e., $x^3y + 4x^2 + xy < 0 \wedge -x^2 + 2xy - 1 < 0 \wedge x < 0$).*

*Effect of different variable orders.* *Figure 1b first eliminates $y$ then $x$ and results in 13 cells, and Figure 1c first eliminates $x$ then $y$ and results in 89 cells, almost seven times that of the former.*

## 2.2 Suggesting Variable Order for Cylindrical Algebraic Decomposition

An **Expert-Based (EB)** heuristic is a sequence of meticulous mechanized rules. It is mainly derived from theoretical analysis or a large number of observations on practical instances and summarized by experts. The heuristics can capture the human-readable characteristics of the problem. A **Learning-Based (LB)** heuristic will suggest an order through the scoring function or a variable selection distribution given by the learning model. It can exploit features deep inside the problem statement via high-dimensional abstraction.

Another important indicator is whether invoking *project*, as the *project* phases are time-consuming for SVO heuristics in practice. In the following, **UP** denotes heuristics utilizing *project*, and **NUP** denotes heuristics not utilizing *project*.

**EB & UP.** The heuristics *sotd* (sum of total degree) [10], and *ndrr* (number of distinct real roots) [12] will project utilizing all different variable orders until the polynomial sets with only one variable. Then *sotd* will select the order with the smallest sum of total degrees for each monomial, while *ndrr* will select the order with the smallest number of distinct real roots. Because of the combinatorial explosion of orders, the heuristics projecting all orders only work on the case with a small number of variables. Based on CAD complexity analysis, *gmods* [13] selects the variable with the lowest degree sum in the polynomial set after each *project* phase.

**EB & NUP.** The heuristics *brown* [9], and *triangular* [11] introduced a series of rules about statistical features like degree, total degree, and occurrence to distinguish the importance of variables. The heuristic *chord* [14] also provides an efficient algorithm based on the associated graph. It makes a variable order via *perfect elimination ordering* on the graph. Note that chord heuristic only works on the *chordal* graph. It is a special case that, after each *project* phase, the graph only removes the linked edges of the projected variable without changing the other components.

**LB & UP.** To the best of our knowledge, no heuristic should be classified into this category.

**LB & NUP.** The approach *EMLP* [15] utilizes an MLP neural network. The network takes the selected statistics of the polynomial set as input and outputs a label for variable order directly. If the

polynomial set has 3 variables, then $3! = 6$ output labels are necessary for the neural network. The approach *PVO* [16] combines neural network and EB & NUP heuristics like *brown* and *triangular*. The neural network is trained to predict the best first variable while the EB & NUP heuristics decide other parts of orders. These kinds of heuristics work on 6 variables at most in their experiments.

According to the classification, our proposed approaches, GRL-SVO(UP) and GRL-SVO(NUP), can be categorized as LB & UP and LB & NUP, respectively.

### 2.3 Graph Neural Network and Reinforcement Learning

Graph Neural Networks (GNNs) are a class of deep learning models for graph-structured data. GNNs include many variants according to the characteristics of the problems, such as GCN [19], GAT [20], superGAT [21], and so on. Reinforcement Learning (RL) is an advanced machine learning paradigm where an agent learns to make decisions by interacting with its environment. It includes various frameworks, such as REINFORCE [22], DQN [23], and so on. By leveraging the expressive power of GNNs to learn complex graph structures and the adaptability of Reinforcement Learning (RL) to find optimal decision-making policies, researchers have remarkably succeeded in combinatorial algorithm design [24, 25, 26]. GRL-SVO is based on an Advantage Actor-Critic (A2C) framework [24, 27] with a Graph Network [28].

## 3 Method

This section starts with the problem formulation for the framework, followed by an overview and description of the graph representation and architecture. Finally, we introduce the state transition without *project*, which is the key technique for GRL-SVO(NUP).

### 3.1 Problem Formulation

We now give the formulation of SVO for CAD. Our goal is to improve CAD efficiency by suggesting a better variable order. Computation time and memory usage are important indicators, but they will be affected by random factors, such as CPU clock, usage rate, etc. As the main output of CAD, cells can be the best candidate. In order to measure the quality of the result, the number of cells is an appropriate indicator that intuitively shows the effect of CAD [29]. In theory, a large number of cells means that the partitions are fragmented compared to a small number of cells. Usually, the polynomial set generated from *project* phase is complex and difficult for the next phases of CAD. In practice, the number of cells also strongly correlates to the computation time and memory usage. Figures of the relation are listed in Appendix B, and we found that the computation time and memory usage increase when the number of cells increases. The objective is to minimize the number of cells $N(Q, \sigma)$, where $Q \subseteq \mathbb{R}[\boldsymbol{x}]$ is a polynomial set with coefficients in $\mathbb{R}$ and $\sigma$ is the given variable order, i.e., $\min N(Q, \sigma)$.

By analyzing the input polynomial set $Q$, we can derive the variable order so that we ought to minimize the objective:
$$\min N(Q, \sigma(Q)).$$

The difficulties of this framework mainly come from two aspects:

- Huge input space. The expression of a polynomial is compressed, and any slight change in form (such as power) will change the geometric space drastically. The EB heuristics may become inefficient when encountering characteristics beyond the summarized patterns.

- Huge output space. The number of variable orders and the number of variables have a factorial relationship, i.e., $n$ variables resulting in $n!$ different variable orders. For example, 10 variables lead to 3628800 candidate variable orders. *sotd*-like, *ndrr*-like, and *EMLP*-like heuristics become impractical due to the vast number of candidate variable orders.

### 3.2 GRL-SVO Overview

In this paper, considering the challenges mentioned above, we propose the GRL-SVO approach, and Figure 2 shows the overall architectures. For huge input space, we compress the polynomial

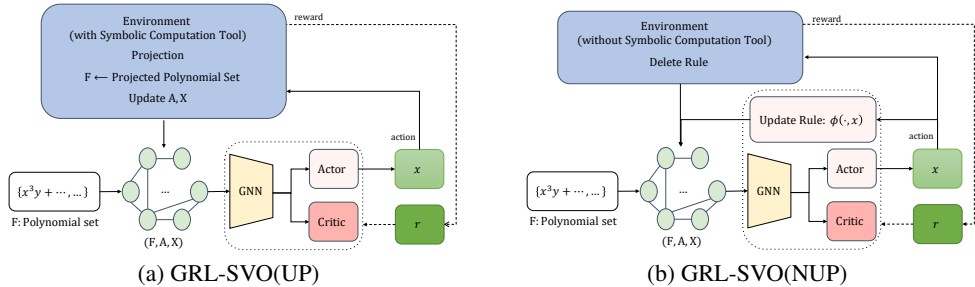

(a) GRL-SVO(UP)            (b) GRL-SVO(NUP)

Figure 2: The architecture of GRL-SVO(UP) and GRL-SVO(NUP) where $\phi(\cdot, x) = MLP(CONCAT(\cdot, x))$ for updating the embedding for the neighbours of $x$. The dashed lines represent that it will be only utilized in training mode.

information into an associated graph [30] with embeddings, which is simple and can depict the relationship between variables. For huge output space, we utilize the neural network to predict the next best variable, and by repeating until no variables are left, the trajectory corresponds to a variable order. In detail, the actor neural network provides a distribution of actions, i.e., the choice of variables. The critic neural network scores the total order and stabilizes the training process as a state-dependent baseline. GNN encodes each variable of the current state as a high-dimensional embedding, and additional neural network components transform them into our policy. As for state transformation, GRL-SVO(UP) and GRL-SVO(NUP) are different in utilizing *project*. The environment of GRL-SVO(UP) projects the selected variable and reorganizes the total graph, while that of GRL-SVO(NUP) updates the graph via the update rule and delete rule.

### 3.3 Graph Representation

The graph of a polynomial set can be different. We introduce a graph structure that can reflect the coupling relationship between variables.

**Definition 2** (Associated Graph [30]). *Given a polynomial set $F$, and the variable set of $F$, $V = var(F)$, an associated graph $G_F(V, E)$ of $F$ is an undirected graph, where $E = \{(x_i, x_j) | \exists f \in F, x_i, x_j \in var(f)\}$.*

In other words, if two variables appear in the same polynomial, they will have an edge.

GNNs have invariance to permutations and awareness of input sparsity [31, 32, 33]. The strength similarly applies to our work. The associated graph is pure and simple, which only retains information related to variables and "neighbors" in the same polynomial. Note that such a graph can easily become a complete graph. For example, $x_1 + x_2 + x_3 + x_4$ corresponds to a complete associated graph. So, we need to distinguish nodes via rich embeddings, detailed in Appendix B. The embeddings are proposed based on former research [16, 34] and our observations. The embedding vectors of variables will be first normalized via the z-score method, i.e.,

$$E'_j[x] = \frac{E_j[x] - mean(\{E_j[v]\})}{std(\{E_j[v]\})}, v \in var(F^0),$$

where $F^0$ is the corresponding polynomial set and $E_j[x]$ denotes the $j$-th scalar of the original embedding of variable $x$.

The graph representation is a tuple $(F^0, A^0, X^0)$ where $A^0 \in \{0, 1\}^{n \times n}$ is the adjacency matrix of the associated graph and $X^0 \in \mathbb{R}^{n \times d}$ is a normalized node embedding matrix for variables.

To encode the representation, we utilize a stack of $k$ GNN layers, Formula (5.7) in [28]. The process encodes the representations $X^{i+1} = [X^{i+1}[0]; \cdots; X^{i+1}[n-1]]$ via

$$X^{i+1}[u] = \sigma(X^i[u] \cdot \theta^i_{g,self} + \sum_{v \in \mathcal{N}(u)} X^i[v] \cdot \theta^i_{g,neigh} + \theta^i_{g,bias}),$$

where $\theta^i_{g,self}$ and $\theta^i_{g,neigh}$ are trainable parameter matrices in $\mathbb{R}^{H^i \times H^{i+1}}$, and $\sigma$ denotes the activation function. $\theta^i_{g,bias}$ is the trainable bias term in $\mathbb{R}^{H^{i+1}}$, and $\mathcal{N}(u)$ is the set of neighbours of variable

$u$. $H^i$ is the dimension of hidden channels, and $H^0 = d$. The layer will aggregate the local information of variables and update the embedding sequentially. Finally, we obtain the intermediate tuple $(F^k, A^k, X^k)$.

## 3.4 Architecture

### 3.4.1 Markov Decision Process (MDP)

**State Space and Action Space.** GRL-SVO(UP/NUP) will suggest a variable order for any polynomial set. The state space includes the graph representation of any polynomial set, i.e., $\mathcal{S} = \{G = (F^0, A^0, X^0)\}$. Although it is a very large space, the state $s$ provides sufficient statistics to evaluate actions. Action corresponds to a candidate variable to *project*. For a given state $s$, the action space is $\mathcal{A} = var(F_s)$, where $F_s$ denotes the polynomial set of current state $s$.

**Environment.** At the time $t$, the environment of GRL-SVO(UP) takes current state $s^t$ and selected variable (action $a^t$) as input and outputs a new state $s^{t+1}$ via processing the projected polynomial set of CAD. That of GRL-SVO(NUP) only removes the selected variable and linked edges from the current state $s^t$ and updates embeddings via neural networks, which is detailed in Section 3.5.

**Reward.** The number of cells is sufficient to reflect the impact of variable order on efficiency. $R(\sigma|s^0) = -N(F_{s^0}, \sigma)/M$ denotes the reward for a given variable order $\sigma$ under the initial state $s^0$, i.e., the negative number of cells divided by a static normalization factor $M$. If the order leads to running timeout and cannot obtain the number of cells, $R(\sigma|s^0) = -1$ directly. The reward of agent policy will increase as the training progresses.

### 3.4.2 Neural Network Architecture

**Neural Network.** The actor network $\theta_a : \mathbb{R}^{n \times d} \to \mathbb{R}^n$ combines an MLP and softmax layer that transforms the $X_t^k$ of $s^t$ to the action distribution at time $t$. The action obeys the distribution, i.e.,

$$a^t \sim \theta_a(X_t^k) = softmax(MLP(X_t^k)).$$

The critic network $\theta_c : \mathbb{R}^{n \times d} \to \mathbb{R}$ is a combination of MeanPool and MLP layers, where $MeanPool : \mathbb{R}^{n \times d} \to \mathbb{R}^d$. The critic value is defined as

$$\theta_c(X_0^k) = MLP(MeanPool(X_0^k)).$$

**Training.** The parameters of GRL-SVO $\theta = \{\theta_g, \theta_a, \theta_c\}$ will go through an end-to-end training process via stochastic gradient descent method. Given initial state $s$, we aim to learn the parameters of a stochastic policy $p_\theta(\sigma|s)$, which assigns high probabilities to order with a small number of cells and low probabilities to order with a large number of cells. Our neural network architecture uses the chain rule to factorize the probability of a variable order $\sigma$ as $p_\theta(\sigma|s) = \prod_{t=1}^n p_\theta(\sigma^t|s, \sigma^{<t})$, where $p_\theta(\sigma^t|s, \sigma^{<t}) = \theta_a(X_t^k)$ is current action distribution. $\sigma^t$ is the $t$-th element in the variable order $\sigma$ and $\sigma^{<t}$ is the partial order from $\sigma^1$ to $\sigma^{t-1}$. The training objective is the expected reward, which is defined as $J(\theta|s) = \mathbb{E}_{\sigma \sim p_\theta(\cdot|s)} R(\sigma|s)$.

Through the well-known REINFORCE algorithm [35], the gradient of the training objective is

$$\nabla_\theta J(\theta|s) = \mathbb{E}_{\sigma \sim p_\theta(\cdot|s)}[(R(\sigma|s) - c(s))\nabla_\theta log p_\theta(\sigma|s)],$$

where $c(s) = \theta_c(X_0^k)$ is the predicting critic value.

Through Monte Carlo sampling, we obtain $N$ $i.i.d$ polynomial sets corresponding to states $s_1, s_2, \cdots, s_N \sim \mathcal{S}$, and $N$ variable orders $\sigma_i \sim p_\theta(\cdot|s_i)$. So we update the parameters of neural networks via

$$\theta_a \leftarrow \theta_a + \frac{1}{N}\sum_{i=1}^N (R(\sigma_i|s_i) - c(s_i))\nabla_\theta log p_\theta(\sigma_i|s_i),$$

$$\theta_c \leftarrow \theta_c + \frac{1}{N}\sum_{i=1}^N (R(\sigma_i|s_i) - c(s_i))\nabla_\theta c(s_i).$$

**Inference.** At inference time, we generate an order via a greedy selection. For the $i$-th element of the order $\sigma$, we select the variable $x$ with the maximal probability, i.e., $x = \arg\max_{x \in V_{t-1}}(p(x|s, \sigma^{<t}))$,

where $V_{t-1} = var(F_{s^{t-1}})$ is the set of variables that have not been projected at time $t$. After selecting variables $n-1$ times, we obtain the total order $\sigma$.

### 3.5 State Transition without *Project*

We provide an LB & NUP heuristic to free from interaction with the symbolic computation tools. It simulates the *project* via a neural network for embedding transformation and a delete rule.

As an example, Figure 3 shows a specific case of state transition. The polynomial set changes after *project* phase the variable $x_4$, where $3x_1^2x_4 - 4x_4^3 - 1$ is reduced to a set $\{x_1-1, x_1+1, x_1^2-x_1+1, x_1^2+x_1+1\}$ without $x_4$ while the polynomial $x_1^3 - 4x_2^2x_3 + 12x_2 + 3$ remains unchanged. So, the embedding of neighbors of $x_4$, i.e., $x_1$, will change greatly while

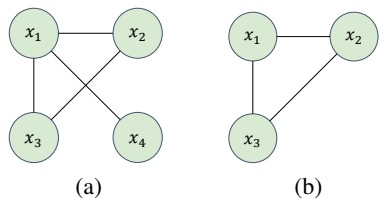

(a)                    (b)

Figure 3: The associated graph of $\{x_1^3 - 4x_2^2x_3 + 12x_2+3, 3x_1^2x_4 - 4x_4^3 - 1\}$ in Figure 3a. Assume that we select $x_4$ to project, it results $\{x_1 - 1, x_1 + 1, x_1^2-x_1+1, x_1^2+x_1+1, x_1^3-4x_2^2x_3+12x_2+3\}$ and the associated graph is shown in Figure 3b.

that of other variables will change slightly. Besides, the projected variable should also be removed from the associated graph. Based on the aforementioned inspiring situations, we propose two rules for approximately simulating *project*.

At time $t-1$, assume $x_i$ is the next projecting variable. We mainly consider the projected variable's influence on its 1-hop neighbor variables.

**Update Rule.** We update the embedding $X$ without *project* for other variables $x_j$ via

$$X[x_j]^t = \begin{cases} \phi(X[x_j]^{t-1}, X[x_i]^{t-1}), & A[x_i][x_j] = 1, \\ X[x_j]^{t-1}, & otherwise, \end{cases}$$

where $\phi(a,b) = MLP(CONCAT(a,b))$ is the neural network that simulates the *project* for embedding transformation.

**Delete Rule.** It will trivially remove $x_i$ and edges linked to $x_i$ and update $A, X$ in the state correspondingly.

$$A \leftarrow RemoveRowColumn(A, Map(x_i)),$$
$$X \leftarrow RemoveRow(X, Map(x_i)),$$
$$Map(x_j) \leftarrow Map(x_j) - 1, i < j < n,$$

where the function $Map(x) : V \rightarrow \mathbb{N}$, maps the variable $x$ to the index in matrix $A$ and $X$, the operation $RemoveRowColumn(A, Map(x_i))$ removes the row and column of variable $x_i$ from $A$, and the operation $RemoveRow(X, Map(x_i))$ removes the row of variable $x_i$ from $X$.

After such transitions, the state will feed the model defined by Section 3.4 and obtain the next projecting variable. After calling the model $n-1$ times, the trajectory corresponds to a variable order suggested by GRL-SVO(NUP).

## 4 Experiments

### 4.1 Setup

**Implementation and Environments.** We utilized PyTorch Geometric [36] for implementations of our approach. The hyper-parameters are listed in Appendix B. We utilized the NVIDIA Tesla V100 GPU for training. After the heuristics output the variable order, all instances with the given variable order are run with MAPLE 2018 on an Intel Xeon Platinum 8153 CPU (2.00GHz). The run-time limit is 900 seconds, and the time to predict a variable order is also counted.

**Dataset.** We utilize two CAD datasets: *random* and *SMT-LIB*. The detailed parameters of *random* generation and collecting methods for *SMT-LIB* are presented in Appendix B.

The *random* dataset contains 7 categories from 3-var to 9-var. We generate 20000 3-var instances and split them into training, testing, and validation sets in a ratio of 8:1:1. We pre-run all $3! = 6$ variable orders and remove non-conforming instances where some variables are eliminated due to random

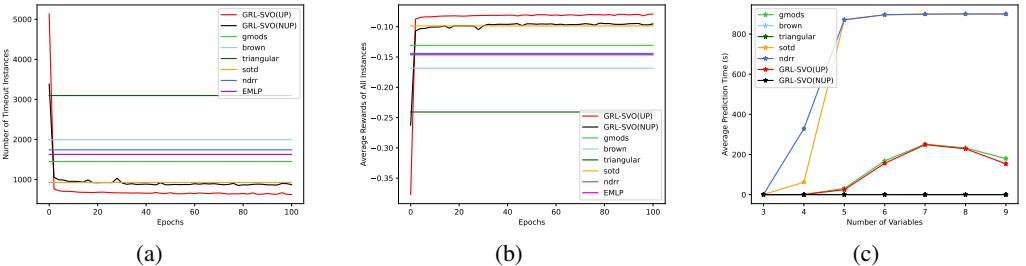

Figure 4: The performance of all heuristics. Figure 4a and 4b correspond to the training phase, and the horizontal lines represent the timeout instances of corresponding heuristics on the training set. Figure 4c is the $PT$ graph over number of variables.

generation, for example, $x - x = 0$; we remove the instances that are all timeout under 6 orders. The rest sets have 14990, 1871, and 1887 instances, respectively. The other categories contain 1000 instances. They are generated by *randpoly* of MAPLE. We also collect a *SMT-LIB* dataset where there are from 3-var to 9-var instances with numbers $\{5908, 1371, 131, 123, 318, 41, 24\}$.

**Baselines.** We compare our approach with two kinds of state-of-the-art heuristics, **UP** or **NUP**. For UP, we compare our approaches with *sotd* and *ndrr* implemented in the ProjectionCAD package [37]. Besides, we also implement *gmods* [13] in MAPLE. For NUP, we compare our approaches with *brown*, and *triangular* implemented in the RegularChains package [38]. *EMLP* approach is proposed by England [15] and its implementation only supports 3-var instances. *PVO(brown)* and *PVO(triangular)* are *PVO* approaches combined with EB & NUP heuristics, *brown* and *triangular*, respectively. The provided implementations [16] only support 4,5,6-var instances.

**Criterion.** Assume $T$ and $N$ denote the running time and number of cells. $AVG.T$ and $AVG.N$ denote the average of $T$ and $N$. If an instance runs timeout, we count the maximum time (900 seconds) and the maximum number of cells of this instance solved by other heuristics into the calculation. We remove the instances that all heuristics lead to CAD timeout, as such instances can not distinguish the ability of heuristics. $PT$ and $\#SI$ denote the variable order prediction time used by heuristics and the number of solved instances within the time limit. For UP heuristics, $PT$ also takes *project* time into consideration. Except for $\#SI$, the criteria are the smaller, the better.

**Experiments.** We conduct three experiments to evaluate our approaches. First of all, we only train the models on the 3-var *random* dataset. Then, we generalize the trained models on *random* datasets with up to 9 variables and the *SMT-LIB* dataset. Finally, we compare GRL-SVO(UP) and GRL-SVO(NUP) and discuss different application scenarios. We list ablation experiments in Appendix C, investigating the effect of features (of the initial embedding), network size, network structure, reward normalization factor $M$, GNN architecture, and coefficient. We also list additional results in Appendix D, including results under other criteria and performance of fine-tuning.

## 4.2 Results

**Training GRL-SVO.** Figure 4a and Figure 4b show the performance during training with the instances in the training set. GRL-SVO outperforms the other heuristics through training and shows a rapid performance improvement. The UP heuristics are better than the NUP heuristics. The inference can be more accurate because the UP heuristics obtain more information than the NUP heuristics. In the beginning, GRL-SVO(NUP) with the random initial parameters is better than GRL-SVO(UP). After training, GRL-SVO(UP) shows better performance.

**Generalization.** We only train on the 3-var dataset and generalize the models to the other datasets. After removing all timeout instances, there are 1876, 651, 416, 354, 349, 409, and 383 instances for the *random* dataset from 3-var to 9-var; for the *SMT-LIB* dataset, 1777, 387, 17 instances for 3-var, 4-var to 6-var and 7-var to 9-var, respectively. Table 1 shows the performance of all heuristics, and the best scores are bolded. GRL-SVO(UP) is the only LB approach with UP heuristics, achieving the best performance among most UP and NUP heuristics except for the 4-var category. GRL-SVO(NUP) also achieves competitive performance compared to other NUP heuristics. GRL-SVO also shows

Table 1: The performance of all heuristics. The dash "-" indicates that the method does not support the category.

| Categories | | NUP | | | | | | UP | | | |
|---|---|---|---|---|---|---|---|---|---|---|---|
| | | EB | | LB | | | | EB | | | LB |
| | | brown | triangular | EMLP | PVO(brown) | PVO(triangular) | GRL-SVO(NUP) | sotd | ndrr | gmods | GRL-SVO(UP) |
| 3-var(test) | #SI | 1620 | 1504 | 1686 | - | - | 1772 | 1784 | 1663 | 1693 | **1798** |
| | AVG.T | 171.41 | 228.32 | 140.87 | - | - | 94.87 | 91.47 | 148.92 | 124.06 | **78.06** |
| | AVG.N | 2427.74 | 2669.67 | 2390.68 | - | - | 2166.67 | 2149.07 | **2007.98** | 2195.80 | 2089.68 |
| 4-var | #SI | 415 | 376 | - | 408 | 392 | 443 | **625** | 488 | 513 | 533 |
| | AVG.T | 352.87 | 394.71 | - | 360.33 | 376.71 | 314.57 | **85.12** | 292.06 | 215.48 | 191.45 |
| | AVG.N | 5241.95 | 5585.90 | - | 5323.83 | 5582.46 | 5131.40 | **3925.28** | 4248.45 | 4849.18 | 4764.50 |
| 5-var | #SI | 236 | 202 | - | 242 | 218 | 238 | 43 | 27 | 329 | **346** |
| | AVG.T | 434.52 | 494.37 | - | 418.34 | 465.43 | 420.51 | 827.47 | 853.75 | 238.01 | **207.79** |
| | AVG.N | 12310.70 | 13224.18 | - | 11795.90 | 12555.82 | 12090.49 | 14538.69 | 14845.67 | 10466.79 | **9744.58** |
| 6-var | #SI | 175 | 149 | - | 180 | 160 | 202 | 5 | 5 | 273 | **306** |
| | AVG.T | 501.75 | 552.14 | - | 490.73 | 527.86 | 439.62 | 889.16 | 889.97 | 284.40 | **214.55** |
| | AVG.N | 20639.07 | 20440.23 | - | 20181.98 | 19290.37 | 19302.97 | 23298.50 | 23329.10 | 17561.67 | **16715.20** |
| 7-var | #SI | 163 | 118 | - | - | - | 153 | 1 | 1 | 270 | **297** |
| | AVG.T | 548.15 | 631.85 | - | - | - | 552.47 | 897.75 | 897.79 | 313.73 | **245.57** |
| | AVG.N | 27790.31 | 27795.79 | - | - | - | 27302.28 | 30452.64 | 30456.31 | 24465.89 | **22432.30** |
| 8-var | #SI | 173 | 138 | - | - | - | 172 | 0 | 0 | 310 | **345** |
| | AVG.T | 601.90 | 654.20 | - | - | - | 597.09 | 900.00 | 900.00 | 372.34 | **322.80** |
| | AVG.N | 39382.26 | 40679.57 | - | - | - | 38815.98 | 43112.02 | 43112.02 | 34016.98 | **33505.21** |
| 9-var | #SI | 151 | 125 | - | - | - | 158 | 0 | 0 | 286 | **325** |
| | AVG.T | 649.41 | 690.29 | - | - | - | 625.69 | 900.00 | 900.00 | 431.11 | **374.78** |
| | AVG.N | 48273.67 | 49832.78 | - | - | - | 46946.09 | 52173.03 | 52173.03 | 42594.25 | **42270.91** |
| SMT-LIB (3-var) | #SI | 1770 | 1763 | 1675 | - | - | 1766 | 1750 | 1694 | **1772** | 1772 |
| | AVG.T | 20.33 | 23.68 | 83.09 | - | - | 22.38 | 34.38 | 65.10 | **18.32** | 18.53 |
| | AVG.N | 4449.79 | 5070.46 | 7661.07 | - | - | 4104.43 | **3672.21** | 4140.72 | 3873.22 | 3946.84 |
| SMT-LIB (4-var to 6-var) | #SI | 374 | 372 | - | 372 | 372 | 364 | 356 | 339 | **379** | 379 |
| | AVG.T | 86.03 | 89.95 | - | 88.32 | 88.98 | 91.17 | 105.18 | 142.32 | **59.96** | 67.51 |
| | AVG.N | 24596.20 | 24260.88 | - | 23039.49 | 22730.34 | 21040.09 | **16896.16** | 21013.25 | 17388.52 | 18894.51 |
| SMT-LIB (7-var to 9-var) | #SI | 13 | 12 | - | - | - | 12 | 11 | 11 | **16** | 14 |
| | AVG.T | 308.14 | 377.32 | - | - | - | 339.90 | 541.53 | 588.33 | **260.53** | 329.91 |
| | AVG.N | 53971.24 | 58675.94 | - | - | - | 51570.88 | 51470.41 | 62185.82 | **50381.12** | 56312.41 |

competitive performance in the real-world dataset, *SMT-LIB* dataset. Note that although the heuristics *sotd* perform better than GRL-SVO(UP) on the 4-var instances, they run timeout in most instances from 5-var due to the combinatorial explosion of the number of variable orders as shown in Figure 4c. The NUP heuristics take a short prediction time, and the polylines in Figure 4c overlap. For example, an instance with 7 variables leads to $7! = 5040$ variable orders to project for *sotd* and *ndrr*.

### 4.3 Discussion on GRL-SVO(UP/NUP)

As in Table 1, GRL-SVO(UP) performs better in $\#SI$, $AVG.T$, $AVG.N$ compared to GRL-SVO(NUP). It is understandable because GRL-SVO(UP) receives more real-world information from the projected polynomial set at each step. As in Figure 4c, GRL-SVO(NUP) is faster than GRL-SVO(UP). The inference time of GRL-SVO(NUP) is almost unchanged with the increase of variables, but there is an obvious increase of GRL-SVO(UP). As the number of variables grows, the time of *project* and interactions between GRL-SVO(UP) and symbolic computation tools will be critical. It is also the bottleneck of UP heuristics like GRL-SVO(UP), *gmods*, *sotd*, and *ndrr*. Therefore, the application scenarios corresponding to the two models will be different.

Internalizing GRL-SVO(UP) into the CAD process is a promising option. As *project* is an algorithm component of CAD, internalization will help reuse the results of projection and reduce the interaction time. GRL-SVO(NUP) is cheap and can extract information directly from polynomial representations. It might be applied to other tools that do not use the entire CAD process. As a canonical example, the automated reasoning tools like Z3 [5], YICES2 [6], CVC5[39], utilize *project* partially only for generating lemmas when solving non-linear arithmetic problems. At the beginning of solving, they also require a fixed variable order. For tasks that are time-critical and do not utilize full CAD in the solving process, GRL-SVO(NUP) seems to be a better option.

## 5 Limitations

Our graph representation cannot embed complete information on polynomials. The associated graph is simple and only shows the relationship between variables. Through selection, we conduct the embedding of graph nodes, but they still ignore plenty of minutiae information, for example, the distribution of coefficients. Besides, the lack of sufficiently large datasets is also a matter of urgency. We can generate large amounts of random data but may lack practical instances for training. Another slight limitation is the prediction time. Python and PyTorch are both heavy techniques. The prediction time of GRL-SVO shown in Figure 4c is actually not much different from other heuristics.

## 6 Conclusion and Future Work

In the paper, we have proposed the first RL-based approach to suggest variable order for cylindrical algebraic decomposition. It has two variants: GRL-SVO(UP) for LB & UP and GRL-SVO(NUP) for LB & NUP. GRL-SVO(UP) can suggest branching variables in the CAD process; GRL-SVO(NUP) can suggest total variable order before the CAD process. Our approaches outperform state-of-the-art learning-based heuristics and are competitive with the best expert-based heuristics. Our RL-based approaches also show a strong learning and generalization ability. Future work is to deploy our approach to practical applications, such as constructing an RL-based package for MAPLE and an algorithm component for automated reasoning tools for non-linear arithmetic solving.

## Acknowledgement

This work has been supported by the National Natural Science Foundation of China (NSFC) under grants No.61972384 and No.62132020. Feifei Ma is also supported by the Youth Innovation Promotion Association CAS under grant No.Y202034. The authors would like to thank Bican Xia, Jiawei Liu, and the anonymous reviewers for their comments and suggestions.

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

# A Cylindrical Algebraic Decomposition

## A.1 Detailed Description

### A.1.1 Polynomial

$\mathbb{N}$ and $\mathbb{R}$ denote the set of natural numbers and real numbers, respectively. Let $\boldsymbol{x} = \{x_1, \cdots, x_n\}$ be the variable set, where $x_1 < x_2 < \cdots < x_n$ is the order of the variables.

A term $t_i$ is a finite production of powers of variables, i.e. $t_i = \prod_{j=1}^{n} x_j^{d_{i,j}}$, where $d_{i,j} \in \mathbb{N}$ as the degree of the variable $x_j$. We denote $\sum_{j=1}^{n} d_{i,j}$ as the *degree* of the term.

A polynomial $P \in \mathbb{R}[\boldsymbol{x}]$ of general form is a finite sum of terms, i.e., $P = \sum_{i=1}^{m} c_i t_i$, where $c_i \in \mathbb{R}$ is coefficient of term $t_i$. In addition, the equivalent nested form of polynomial $Q \in \mathbb{R}[x_1, \cdots, x_{i-1}, x_i]$,

$$Q = a_m x_i^{d_m} + a_{m-1} x_i^{d_{m-1}} + \cdots + a_0,$$

where $0 < d_1 < \cdots < d_m$, and the coefficients $a_i$ are polynomials in $\mathbb{R}[x_1, \cdots, x_{i-1}]$ with $a_m \neq 0$. We denote $x_i$ as *main variable*, $d_m$ as *degree*, $a_m$ as *leading coefficient* of the polynomial $Q$. For example, given a variable order $x_1 < x_2 < x_3$,

$$\begin{aligned} p(x_1, x_2, x_3) &= x_1 x_2 x_3^2 + x_2 x_3^2 + x_3^2 + x_1 x_3 + x_2 x_3 \\ &= ((x_1 + 1)x_2)x_3^2 + (x_1 + x_2)x_3. \end{aligned}$$

The degree of $x_1 x_2 x_3^2$, $x_2 x_3^2$, $x_3^2$, $x_1 x_3$, $x_2 x_3$ are 4,3,2,2,2, while that of the polynomial $p(x_1, x_2, x_3)$ is 2, and leading coefficient is $((x_1 + 1)x_2)$.

### A.1.2 Cylindrical Algebraic Decomposition

A Cylindrical Algebraic Decomposition (CAD) is a decomposition algorithm of a set of polynomials in ordered $\mathbb{R}^n$ space resulting in finite sign-invariant regions, named *cells*. After CAD, it can query a limited set of sample points in corresponding cells. Due to the sign of each polynomial is either always positive, always negative, or always zero on any given cell, one can determine the sign of the polynomials at any point in $\mathbb{R}^n$ using this set of sample points.

Note that computing the CAD of a set of univariate polynomials (one-dimensional CAD) is quite simple. We only need to calculate all the real roots of the polynomials, and the cells are precisely these real roots themselves along with the intervals they divide. This leads to a motivation for computing the CAD of a set of polynomials with $n$ variables, which involves recursively reducing the construction of a $k$-dimensional CAD to the construction of a $(k-1)$-dimensional CAD, until reaching the recursive boundary of computing a one-dimensional CAD, and then constructing higher-dimensional CAD from lower-dimensional CAD continuously.

Formally, the algorithm consists of three components: projection, root isolation, and lift. A diagram is shown in Figure 5. The *project* phase eliminates the variables of a polynomial set $P_n$ with $n$ variables by a strictly defined projection operator *proj* in a given order $x_1 < x_2 < \cdots < x_n$ until only one variable $x_1$ left, resulting in polynomial sets $P_1, \cdots, P_{n-1}$, where $P_i = proj(P_{i+1})$ contains only the variables $x_1, \cdots, x_i$. The projection operator is carefully designed to ensure that the CAD of $P_i$ can be constructed from the CAD of $P_{i-1}$. Then the *root isolate* and *lift* phases are alternated successively. To make the following statement compatible, we replace the notation of $P_1$ with $P_{1,1}$. Let $N(i)$ denote the number of cells generated by $P_i$ (also equals to the number of sample points). We set $N(0)$ to be 1. The *root isolate* phase will output all roots of a univariate polynomial set. When $i = 1$, the roots of $P_{1,1}$ split $\mathbb{R}$ into $N(1)$ cells. Let $SP_{i,k}, 1 \leq k \leq N(i-1)$ denote the set of sample points of the cells generated by $P_{i,k}$ and $SP_i$ denote the union of $SP_{i,k}$. Actually, $SP_i$ is precisely the sample points of the cells generated by $P_i$ and $|SP_i| = N(i)$. The *lift* phase assigns each sample point $s_{i,k}$ of $SP_i$ to variables of $P_{i+1}$, i.e., $(x_1, x_2, \cdots, x_i) \leftarrow s_{i,k}, 1 \leq k \leq N(i)$, to polynomials in $P_{i+1}$, resulting in univariate polynomial sets $P_{i+1,k}, 1 \leq k \leq N(i)$. Invoking the *root isolate* phase will obtain each $SP_{i+1,k}$ and finally their union $SP_{i+1}$. After repeating *root isolate* and *lift* for $n-1$ times, we achieve the cells of $P_n$ characterized by the sample points $SP_n$. The paper provides a concrete example of the CAD process in Example 2.1.

The projection operator *proj* plays a key role in the CAD process, which carries enough information to ensure that the CAD of any set of polynomials $P$ can be constructed from the CAD of $proj(P)$.

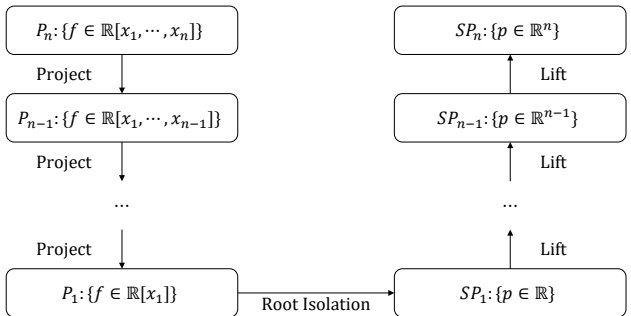

Figure 5: The process of CAD.

The first projection operator that satisfies the above property is designed by George Collins[2], which is too complicated, however. Here, we introduce another classic simplified projection operator, the McCallum Projection Operator[40]. It is the default projection operator used in the ProjectionCAD package[37] and our architecture of GRL-SVO(UP).

In mathematics, the resultant is used to determine whether two polynomials have common zeros, while the discriminant is used to determine if one polynomial has repeated roots. Both these two tools are crucial in the construction of the projection operator. We first introduce the definitions of the resultant and the discriminant, and then the McCallum Projection Operator is detailed in Definition 5.

**Definition 3** (Resultant). *Let $f_1$, $f_2$ be two polynomials in $\mathbb{R}[x_1, \ldots, x_n]$. Assume that*

$$f_1 = a_m x_n^{d_m} + a_{m-1} x_n^{d_{m-1}} + \cdots + a_0,$$

$$f_2 = b_n x_n^{d_n} + b_{n-1} x_n^{d_{n-1}} + \cdots + b_0.$$

*The resultant of $f_1$ and $f_2$ with respect to $x_n$, $Res(f_1, f_2, x_n)$, is:*

$$Res(f_1, f_2, x_n) = \begin{vmatrix} a_m & a_{m-1} & \cdots & a_0 & & & \\ & a_m & a_{m-1} & \cdots & a_0 & & \\ & & \ddots & \ddots & \ddots & \ddots & \\ & & & a_m & a_{m-1} & \cdots & a_0 \\ b_n & b_{n-1} & \cdots & b_0 & & & \\ & b_n & b_{n-1} & \cdots & b_0 & & \\ & & \ddots & \ddots & \ddots & \ddots & \\ & & & b_n & b_{n-1} & \cdots & b_0 \end{vmatrix}$$

**Definition 4** (Discriminant). *Let $f$ be a polynomial in $\mathbb{R}[x_1, \ldots, x_n]$. Assume that*

$$f = a_m x_n^{d_m} + a_{m-1} x_n^{d_{m-1}} + \cdots + a_0.$$

*The discriminant of $f$ with respect to $x_n$, $Dis(f, x_n)$, is:*

$$Dis(f, x_n) = \frac{(-1)^{\frac{m(m-1)}{2}}}{a_m} Res(f, \frac{\partial f}{\partial x_n}, x_n).$$

**Definition 5** (McCallum Projection Operator). *Let $F = \{f_1, \ldots, f_k\}$ be a set of polynomials in $\mathbb{R}[x_1, \ldots, x_n]$. The McCallum projection operator, $proj_m$, is a mapping that maps $F$ to $proj_m(F)$, where $proj_m(F)$ is the set of polynomials in $\mathbb{R}[x_1, \ldots, x_{n-1}]$ defined by:*

- *The coefficients of each polynomial in $F$,*

- *The discriminant of each polynomial in $F$ with respect to $x_n$,*

- *The resultant of any two different polynomials $f_i$, $f_j$ in $F$ with respect to $x_n$.*

Another important component in the CAD algorithm is the real root isolation algorithm, which accepts a set of univariate polynomials and results in all the roots (actually, the arbitrarily small intervals that contain each root) of the univariate polynomials. This can be accomplished by invoking

the subalgorithm *RootIsolation* multiple times and adjusting the upper and lower bounds of the initial interval for each univariate polynomial.

We provide the specifics of the algorithm *RootIsolation* in Algorithm 2. Before that, some necessary concepts like the sign variation, the 1-norm, and the Sturm sequence are introduced, which play a role in the algorithm *RootIsolation*.

**Definition 6** (Sign Variation). *For a sequence of non-zero real numbers $\overline{c}$: $c_1$, $c_2$, $\cdots$, $c_k$, the sign variation of $\overline{c}$, $V(\overline{c})$, is:*

$$V(\overline{c}) = |\{i|1 \le i < k \& c_i c_{i+1} < 0\}|.$$

*For a sequence of univariate polynomials $\overline{S}$: $f_1$, $f_2$, $\cdots$, $f_k$, the sign variation of $\overline{S}$ at real number $a$, $V_a(\overline{S})$ is:*

$$V_a(\overline{S}) = V(\overline{s}),$$

*where $\overline{s}$ is the real number sequence $f_1(a)$, $f_2(a)$, $\cdots$, $f_k(a)$.*

**Definition 7** (1-norm). *Given a univariate polynomial $f = a_m x^{d_m} + a_{m-1} x^{d_{m-1}} + \cdots + a_0$. The 1-norm of $f$, $||f||_1$, is:*

$$||f||_1 = \sum_{i=0}^{m} |a_i|.$$

---

**Algorithm 1** SturmSequence

**Input** : $f$: a univariate polynomial
**Output**: $sturm$: the Sturm Sequence of $f$

1: $sturm \leftarrow []$
2: $h \leftarrow f$
3: $g \leftarrow f'$
4: $r \leftarrow -rem(g, h, x)$
5: **while** $r \ne 0$ **do**
6:     Append $r$ to $sturm$
7:     $h \leftarrow g$
8:     $g \leftarrow r$
9:     $r \leftarrow -rem(g, h, x)$
10: **end while**
11: **return** $sturm$

---

**Algorithm 2** RootIsolation

**Input** : $f$: a univariate polynomial
**Output**: $(a, b)$: $f$ has a real root in $(a, b)$ where $a, b$ are rational numbers

1: $\overline{S} \leftarrow SturmSequence(f)$
2: $a \leftarrow -||f||_1$
3: $b \leftarrow ||f||_1$
4: **if** $V_a(\overline{S}) = V_b(\overline{S})$ **then**
5:     **return** failure
6: **end if**
7: **while** $V_a(\overline{S}) - V_b(\overline{S}) > 1$ **do**
8:     $c \leftarrow \frac{a+b}{2}$
9:     **if** $V_a(\overline{S}) > V_c(\overline{S})$ **then**
10:         $b \leftarrow c$
11:     **else**
12:         $a \leftarrow c$
13:     **end if**
14: **end while**
15: **return** $(a, b)$

---

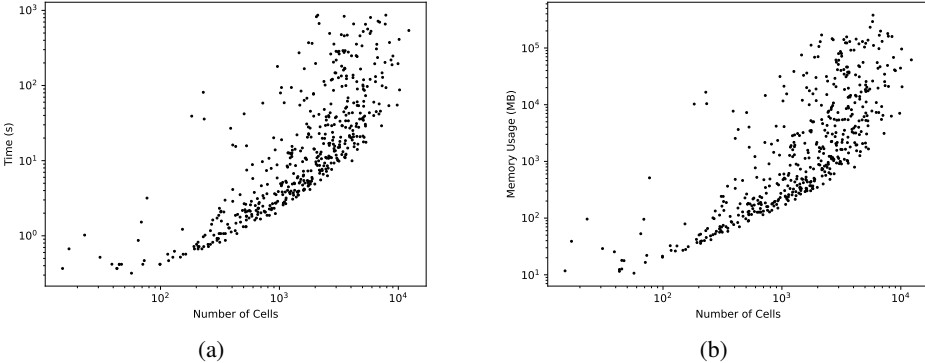

(a)                                  (b)

Figure 6: The relationships between the number of cells and other important indicators. Figure 6a and Figure 6b respectively correspond to the relationships between the number of cells and computation time and the relationships between the number of cells and memory usage. Five hundred instances of 3-var are randomly selected from the *random* dataset to construct the two scatterplots.

## A.2   Case Study: Discriminant

Actually, whether $x^2 + bx + c = 0$ has a real root is equivalent to checking the satisfiability of $\exists x . x^2 + bx + c = 0$. Let's elaborate on why the quantified formula $\exists x . x^2 + bx + c = 0$ can be transformed into an equivalent quantifier-free formula $b^2 - 4c \geq 0$ via CAD techniques. By feeding the CAD algorithm with the set of polynomial $\{x^2 + bx + c\}$ and an order $b \prec c \prec x$, CAD decomposes $\mathbb{R}^3$ into 9 cells:

$$\begin{cases} b = b \\ c < \frac{b^2}{4} \\ x < -\frac{b}{2} - \frac{\sqrt{b^2-4ac}}{2}, \end{cases} \quad \begin{cases} b = b \\ c < \frac{b^2}{4} \\ x = -\frac{b}{2} - \frac{\sqrt{b^2-4ac}}{2}, \end{cases} \quad \begin{cases} b = b \\ c < \frac{b^2}{4} \\ x = -\frac{b}{2} + \frac{\sqrt{b^2-4ac}}{2}, \end{cases}$$

$$\begin{cases} b = b \\ c < \frac{b^2}{4} \\ x > -\frac{b}{2} + \frac{\sqrt{b^2-4ac}}{2}, \end{cases} \quad \begin{cases} b = b \\ c < \frac{b^2}{4} \\ -\frac{b}{2} - \frac{\sqrt{b^2-4ac}}{2} < x < -\frac{b}{2} + \frac{\sqrt{b^2-4ac}}{2}, \end{cases}$$

$$\begin{cases} b = b \\ c = \frac{b^2}{4} \\ x < -\frac{b}{2}, \end{cases} \quad \begin{cases} b = b \\ c = \frac{b^2}{4} \\ x = -\frac{b}{2}, \end{cases} \quad \begin{cases} b = b \\ c = \frac{b^2}{4} \\ x > -\frac{b}{2}, \end{cases} \quad \begin{cases} b = b \\ c > \frac{b^2}{4} \\ x = x. \end{cases}$$

Note that the sign of the polynomial $x^2 + bx + c$ is zero if and only if $(b, c, x)$ belongs to cells:

$$\begin{cases} b = b \\ c = \frac{b^2}{4} \\ x = -\frac{b}{2}, \end{cases} \quad \begin{cases} b = b \\ c < \frac{b^2}{4} \\ x = -\frac{b}{2} - \frac{\sqrt{b^2-4ac}}{2}, \end{cases} \quad \begin{cases} b = b \\ c < \frac{b^2}{4} \\ x = -\frac{b}{2} + \frac{\sqrt{b^2-4ac}}{2}. \end{cases}$$

So we know that there must exist $x$ such that $x^2 + bx + c = 0$ when $b^2 - 4c > 0$ or $b^2 - 4c = 0$, and it is impossible to find a $x$ such that $x^2 + bx + c = 0$ when $b^2 - 4c < 0$. So, we can conclude the quantified formula $\exists x . x^2 + bx + c = 0$ is equivalent to the quantifier-free formula $b^2 - 4c \geq 0$. In fact, similar processes can be abstracted into a universal algorithm to solve the quantifier elimination problems in the real closed field. See [2] for more details.

## A.3   Relation of #Cells And Efficiency

As in Figure 6, there is a strong correlation between the number of cells produced and the computation time, as well as the memory usage. When the number of cells increases, the computation time and the memory usage also increase.

| Table 2: Parameters of *randpoly* function. | |
|---|---|
| **Parameter** | **Description** |
| vars | List or set of variables |
| coeffs | Generator of coefficients |
| expons | Generator of exponents |
| terms | Number of terms |
| degree | Total degree for a dense polynomial |
| dense | The polynomial is to be dense |
| homogeneous | The polynomial is to be homogeneous |

Table 3: Parameters of *random* dataset generation.

| #vars | coeffs | expons | terms |
|---|---|---|---|
| 3 | rand(-100..100) | rand(0..2) | rand(3..6) |
| 4 | rand(-100..100) | rand(0..2) | 3 |
| 5 | rand(-100..100) | rand(0..2) | 3 |
| 6 | rand(-100..100) | rand(0..2) | 3 |
| 7 | rand(-100..100) | rand(0..2) | 3 |
| 8 | rand(-100..100) | rand(0..2) | 3 |
| 9 | rand(-100..100) | rand(0..2) | 3 |

# B Experiment Setup

## B.1 Datasets

### B.1.1 *Random* Dataset

We generated the random polynomial set via the *randpoly(vars, opts)* function in MAPLE, where *opts* are specifying properties like *coeffs, expons, terms, degree, dense, homogeneous*. Table 2 lists the descriptions of parameters. We also show an example of random polynomial generation.

For example, *randpoly([$x_1, x_2, x_3$], terms = 4, expons = rand(0..2), coeffs = rand(-100..100))* will generate a polynomial,

$$56x_1x_2^2x_3^2 - 4x_1^2x_2x_3 + 37x_1x_2^2x_3 - 32x_1x_2x_3^2,$$

where *rand(a..b)* is a random number generator in the range of [*a, b*]. Note that the random polynomial is *sparse* and *non-homogeneous* by default. We also ignore the parameter *degree*, because it is only valid in the case of *dense* random polynomial generation.

Table 3 lists the parameters for *randpoly* for generating the *random* dataset, where *#vars = n* corresponds to a list of variables $[x_1, x_2, \cdots, x_n]$. The random number for *terms* is generated with Python's *random* library outside the MAPLE script. If the polynomial produced by MAPLE lacks a constant term, one is added, ranging from -100 to 100, using the same Python library.

### B.1.2 *SMT-LIB* Dataset

The *SMT-LIB* dataset is built by the instances that only involve real constraints in the QF_NRA category of the *SMT-LIB* [17]. Using the Python interface of Z3[5], we parse the instances to extract the polynomial sets, discarding any instances that cannot be correctly parsed. Then we categorize the processed polynomial sets based on the number of variables and finally build the *SMT-LIB* dataset, containing instances ranging from 3 to 9 variables with counts $\{5908, 1371, 131, 123, 318, 41, 24\}$. Furthermore, we observed that there are numerous polynomial sets in the dataset that yield the same result after factoring out the irreducible factors, which are indistinguishable from CAD algorithms. So we cluster them and only one instance of each clustering is included in the dataset, resulting in $\{1777, 387, 17\}$ instances for 3-var, 4-var to 6-var and 7-var to 9-var, respectively.

## B.2 Neural Networks

Table 4 lists the hyper-parameters of the GNN encoder, the networks used in GRL-SVO(NUP)'s architecture to transform the original embeddings linearly and simulate the *project* process, actor network, critic network, and RL architecture.

# C Ablation Experiments

## C.1 The Effect of Features

There are 14 different features to characterize a variable listed in Table 5. We conduct an experiment on the effect of features. We make masks for these features, where the mask will set the features that we do not care about as zero.

Table 4: The hyper-parameters.

| Category | Parameter | Value |
|---|---|---|
| GNN | The number of GNN layers | 4 |
| GNN | The number of Intermediate layer features | 256 |
| GNN | Aggregation method | mean |
| GNN | Use bias | True |
| NUP_transform | The size of MLP layer features | [14, 256, 128, 64] |
| NUP_simulate | The size of MLP layer features | [128, 512, 256, 64] |
| Actor | The size of MLP layer features | [256, 512, 128, 1] |
| Critic | The size of MLP layer features | [256, 512, 128, 1] |
| RL | Batch size | 32 |
| RL | Learning rate | 2e-5 |
| RL | Training maximum epoch | 100 |
| RL | Reward normalization factor ($M$) | 50000 |

Table 5: The original embedding of a variable in an associated graph.

| Symbol | Description |
|---|---|
| $E_1(x)$ | Number of other variables occurring in the same polynomials |
| $E_2(x)$ | Number of polynomials containing $x$ |
| $E_3(x)$ | Maximum degree of $x$ among all polynomials |
| $E_4(x)$ | Sum of degree of $x$ among all polynomials |
| $E_5(x)$ | Maximum degree of leading coefficient of $x$ among all polynomials |
| $E_6(x)$ | Maximum number of terms containing $x$ among all polynomials |
| $E_7(x)$ | Maximum degree of all terms containing $x$ |
| $E_8(x)$ | Sum of degree of all terms containing $x$ |
| $E_9(x)$ | Sum of degree of leading coefficient of $x$ |
| $E_{10}(x)$ | Sum of number of terms containing $x$ |
| $E_{11}(x)$ | Proportion of $x$ occurring in polynomials |
| $E_{12}(x)$ | Proportion of $x$ occurring in terms |
| $E_{13}(x)$ | Maximum number of other variables occurring in the same term |
| $E_{14}(x)$ | Maximum number of other variables occurring in the same polynomial |

1. One-hot masks (test the effect of a single feature), for example, to test the effect of $E_1$, the corresponding one-hot mask is $(1, 0, 0, 0, 0, 0, 0, 0, 0, 0, 0, 0, 0, 0)$. Multiplying with input feature will result in a feature vector with only $E_1$ while others are 0.

2. Operation masks (test the effect of different operations in features) will group features according to their operation type (maximum, sum, and proportion). Note that we treat $E_1, E_2$ as sub-features utilizing sum operation.

   - Max: $E_3, E_5, E_6, E_7, E_{13}, E_{14}$
   - Sum: $E_1, E_2, E_4, E_8, E_9, E_{10}$
   - Prop: $E_{11}, E_{12}$

3. Object masks (test the effect of different objects in features) will group features according to their target objects (variable, term, and polynomials).

   - Var: $E_1, E_3, E_4, E_{13}, E_{14}$
   - Term: $E_5, E_6, E_7, E_8, E_9, E_{10}, E_{12}$
   - Poly: $E_2, E_{11}$

4. Because degree is a common feature that most heuristics are concerned with, testing the effect of degree is necessary. Degree masks will group features according to whether they utilize degree.

   - Degree: $E_3, E_4, E_5, E_7, E_8, E_9$
   - NoDegree: $E_1, E_2, E_6, E_{10}, E_{11}, E_{12}, E_{13}, E_{14}$

Table 6 and Table 7 show the results of different single, operation, object, and degree features. We can conclude that only one feature is not enough. The sum, term, and degree may be the most important factors, as using Sum/Term/Degree features will result in the largest difference in performance.

Table 6: The performance of different single features.

| | | $E_1$ | $E_2$ | $E_3$ | $E_4$ | $E_5$ | $E_6$ | $E_7$ | $E_8$ | $E_9$ | $E_{10}$ | $E_{11}$ | $E_{12}$ | $E_{13}$ | $E_{14}$ |
|---|---|---|---|---|---|---|---|---|---|---|---|---|---|---|---|
| | #SI | 1459 | 1458 | 1467 | 1635 | 1560 | 1589 | 1461 | 1647 | 1613 | **1670** | 1462 | **1670** | 1459 | 1459 |
| NUP | $AVG.T$ | 252.02 | 250.85 | 247.38 | 151.71 | 198.43 | 186.27 | 251.03 | 157.77 | 175.60 | **147.61** | 249.17 | **147.61** | 252.02 | 252.02 |
| | $AVG.N$ | 2874.94 | 2859.65 | 2852.56 | **2246.06** | 2642.66 | 2592.94 | 2861.09 | 2478.10 | 2515.79 | 2508.01 | 2857.07 | 2508.04 | 2874.94 | 2874.97 |
| | #SI | 1459 | 1540 | 1596 | 1693 | 1632 | 1627 | 1461 | 1687 | 1667 | **1706** | 1540 | **1706** | 1459 | 1459 |
| UP | $AVG.T$ | 252.33 | 211.51 | 184.73 | **121.36** | 163.03 | 165.65 | 251.22 | 137.31 | 145.16 | 129.87 | 211.53 | 129.86 | 252.23 | 252.24 |
| | $AVG.N$ | 2874.97 | 2743.79 | 2755.63 | **2174.76** | 2559.63 | 2531.44 | 2860.81 | 2394.39 | 2465.49 | 2386.24 | 2743.79 | 2386.24 | 2874.87 | 2874.90 |

Table 7: The performance of different feature classifications.

| | | Operation | | | Object | | | Degree | |
|---|---|---|---|---|---|---|---|---|---|
| | | Max | Sum | Prop | Var | Term | Poly | Degree | NoDegree |
| | #SI | 1640 | **1754** | 1670 | 1635 | 1726 | 1462 | 1723 | 1685 |
| NUP | $AVG.T$ | 159.42 | **101.21** | 147.48 | 152.11 | 116.39 | 249.17 | 114.95 | 141.65 |
| | $AVG.N$ | 2444.09 | **2145.71** | 2496.74 | 2250.21 | 2300.32 | 2857.07 | 2177.08 | 2468.04 |
| | #SI | 1716 | **1788** | 1707 | 1698 | 1751 | 1540 | 1770 | 1715 |
| UP | $AVG.T$ | 119.80 | **81.93** | 129.53 | 119.32 | 102.07 | 211.55 | 91.31 | 125.53 |
| | $AVG.N$ | 2346.79 | **2082.22** | 2369.74 | 2184.08 | 2232.74 | 2743.79 | 2103.21 | 2359.62 |

## C.2 The Effect of Network Size

We conduct an experiment on the effect of network size, and the results are listed in Table 8. The number of GNN layers (for short, #G) $\in \{1, 2, 3, 4, 5\}$ and the number of Intermediate layer features (for short, #I) $\in \{32, 64, 128, 256, 512\}$ where Actor and Critic will keep the same proportion 2:4:1. Note that the input dimension of Actor and Critic is the same as #I. For example, if #I = 32, then the dimension of the Actor and Critic are both [32, 64, 16, 1]. Elements in the following tables are #SI in the validation set and #SI in the testing set, respectively.

## C.3 The Effect of Network Structure

We conduct an experiment on the effect of network structure. We build two models: one without embedding (NO_EMB), and the other without edge (NO_EDGE). Note that the GNN updating operator we used is $\mathbf{x}'_i = \mathbf{W}_1 \mathbf{x}_i + \mathbf{W}_2 \sum_{j \in \mathcal{N}(i)} e_{j,i} \cdot \mathbf{x}_j + \mathbf{b}$. As edges are not considered in NO_EDGE, actually, the operator will be $\mathbf{x}'_i = \mathbf{W}_1 \mathbf{x}_i + \mathbf{b}$. So NO_EDGE is equivalent to MLP. For NO_EMB, $\#SI = 1459, AVG.T = 252.03, AVG.N = 2874.91$. Table 9 shows the result of NO_EDGE on testing set. The performance of NO_EMB drops dramatically while that of NO_EDGE is good, and GRL-SVO can still outperform such models. We explore the performance of NO_EDGE(NUP) under different sizes of parameters with GRL-SVO(NUP). MLP_4_512 has twice as many parameters as ours. GRL-SVO(NUP) can outperform all MLPs as shown in Figure 7. It is the advantage of the graph structure where a variable can grasp neighbor information.

## C.4 The Effect of Reward Normalization Factor ($M$)

We make an ablation experiment on $M$ that we train the models with $M$=10000, 20000, 50000(ours), 100000, and without $M$. Note that if there is no $M$, the reward (the number of cells) will be a relatively large integer. As shown in Table 10, $M$ is necessary. The first number is the result of the validation set, while the second is the result of the testing set.

## C.5 The Effect of GNN Architecture

We conduct experiments using different GNN architectures available in PyTorch Geometric that have similar formal parameters as the GNN architecture we used: ClusterGCNConv[41], EGConv[42],

Table 8: The performance (#SI) of GRL-SVO(NUP) and GRL-SVO(UP).

| | GRL-SVO(NUP) | | | | | GRL-SVO(UP) | | | | |
|---|---|---|---|---|---|---|---|---|---|---|
| | 32 | 64 | 128 | 256 | 512 | 32 | 64 | 128 | 256 | 512 |
| 1 | 1754,1737 | 1756,1741 | **1771**,1765 | 1770,1765 | 1764,1765 | 1767,1759 | 1768,1758 | 1777,1774 | 1789,1789 | 1790,1793 |
| 2 | **1771**,1764 | 1762,1747 | 1765,1763 | 1767,1766 | 1764,1768 | 1773,1763 | 1780,1777 | 1778,1776 | 1786,1793 | 1788,**1794** |
| 3 | 1765,1758 | 1756,1744 | 1766,1758 | 1761,1766 | 1770,**1769** | 1781,1767 | 1783,1778 | 1782,1785 | 1788,**1794** | 1786,1793 |
| 4 | 1765,1761 | 1768,1761 | 1764,1762 | 1766,1765 | 1769,1765 | 1775,1766 | 1784,1775 | 1780,1779 | **1792**,**1794** | 1788,1793 |
| 5 | 1768,1762 | 1765,1763 | **1771**,1759 | 1766,1761 | 1763,1762 | 1779,1779 | 1784,1781 | 1788,1797 | **1792**,1790 | 1791,1790 |

Table 9: The performance of NO_EDGE under different sizes.

|  | MLP_2_256 | MLP_3_256 | MLP_4_256 | MLP_4_512 |
|---|---|---|---|---|
| #SI | 1763 | 1763 | 1764 | 1756 |
| AVG.T | 97.89 | 98.43 | 97.68 | 99.08 |
| AVG.N | 2132.33 | 2140.18 | 2129.33 | 2148.64 |

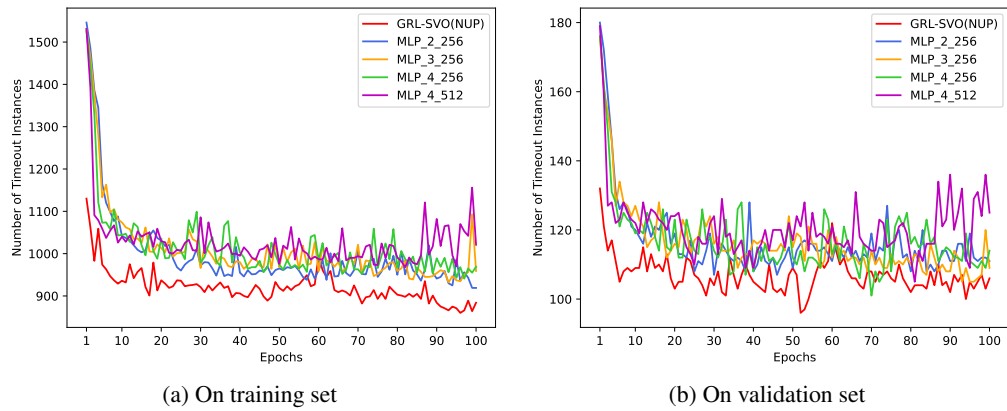

(a) On training set  (b) On validation set

Figure 7: The training process of NO_EDGE and GRL-SVO(NUP).

FiLmConv[42], LEConv[42], GATConv [20], GATv2Conv [43], GeneralConv [44], ResGatedGraph-Conv [45], SageConv [46], TransformerConv [47]. We could observe from Table 11 that each graph neural network (GNN) is capable of effectively learning this problem. Our approach is not reliant on the specific GNN structure.

## C.6  The Effect of Coefficient

With regard to the choice of variable order, the current works do not consider coefficient (both Expert-Based or Learning-Based heuristics). Some experts in Symbolic Computation believe that the reason may be related to the calculation of CAD projection: CAD projection uses two polynomials to make a resultant (Definition 3) to eliminate a common variable, so it first depends on the common variable set of these two polynomials (which does not involve coefficient); secondly, the amount of calculation generally depends on the degree of public variables because the degree determines the size of the resultant matrix. For practical instances, the variation range of the coefficients is uncontrollable, which will increase the difficulty of Learning-Based design and training the neural network. The impact of the coefficient on the problem is complex. We provide some intuition through Figure 8.

(a) $\{x^3y + 4x^2 + xy, -x^2 + 2xy - 1\}, x \prec y : 13, y \prec x : 89;$

(b) $\{x^3y + 8x^2 + xy, -x^2 + 2xy - 1\}, x \prec y : 13, y \prec x : 89;$

(c) $\{x^3y - 4x^2 + xy, -x^2 + 2xy - 1\}, x \prec y : 45, y \prec x : 125;$

(d) $\{x^3y + 4x^2 + xy, x^2 + 2xy - 1\}, x \prec y : 29, y \prec x : 101;$

(e) $\{-x^3y + 4x^2 + xy, -x^2 + 2xy - 1\}, x \prec y : 45, y \prec x : 97.$

Table 10: The performance of different $M$.

|  |  | NO_M | 1000 | 10000 | 50000 (ours) | 100000 |
|---|---|---|---|---|---|---|
| NUP | $\#SI$ | 1677, 1666 | 1672, 1669 | 1764, **1769** | **1766**, 1765 | **1766**, 1765 |
|  | $AVG.T$ | 133.06, 142.19 | 140.28, 142.56 | 93.03, **95.57** | **91.68**, 97.78 | 93.46, 97.51 |
|  | $AVG.N$ | 2230.65, 2214.20 | 2250.11, 2222.47 | 2134.50, **2136.30** | **2129.45**, 2142.98 | 2147.43, 2157.56 |
| UP | $\#SI$ | 1182, 1196 | 1726, 1723 | 1789, **1795** | **1792**, **1795** | 1791, 1793 |
|  | $AVG.T$ | 387.16, 381.11 | 109.12, 115.13 | 73.86, 80.20 | **72.02**, **79.57** | 73.68, 79.86 |
|  | $AVG.N$ | 3125.50, 3101.32 | 2100.53, 2097.48 | 2058.08, 2081.92 | 2062.09, 2075.23 | **2058.05**, **2070.75** |

Table 11: The performance of GRL-SVO(NUP) with different GNN architectures.

|  | ClusterGCNConv | EGConv | FiLmConv | LEConv | GATConv |
|---|---|---|---|---|---|
| #SI | 1767 | 1765 | 1760 | 1761 | 1726 |
| AVG.T | 94.26 | 98.53 | 96.49 | 96.76 | 119.72 |
| AVG.N | 2144.66 | 2177.32 | 2136.67 | 2151.75 | 2166.78 |
|  | GATv2Conv | GeneralConv | ResGatedGraphConv | SageConv | TransformerConv |
| #SI | 1762 | 1768 | 1759 | 1765 | 1769 |
| AVG.T | 104.14 | 95.17 | 96.25 | 96.74 | 94.45 |
| AVG.N | 2184.86 | 2132.81 | 2146.59 | 2131.22 | 2134.51 |

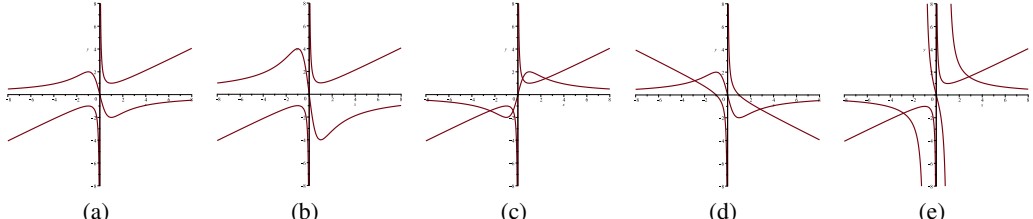

| (a) | (b) | (c) | (d) | (e) |

Figure 8: The results of slight changes of coefficients.

The number of cells of (a) and (b) are the same, while (c), (d), and (e) are different. But the best variable order is the same ($x \prec y$) in these cases. To a certain extent, in these cases, the coefficient mostly affects the number of cells. We conduct an experiment on coefficient that we have randomly modified the coefficients (in [-100, 100]) of 1000 instances randomly selected from a 3-var testing set. Since the coefficients were the only modification made, we used the original variable order generated from the unaltered instances. Our models continue to outperform other heuristics, as demonstrated by Table 12. We observe a slight decline in the performance of all heuristics, indicating that the coefficient also plays a significant role as a parameter (although it may not be the most crucial one).

# D    Additional Results

## D.1    Results Under Other Criteria

Assume $T$ and $N$ denote the running time and number of cells. $COMAVG.T$ and $COMAVG.N$ denote the average of $T$ and $N$ of the instances that all heuristics solved within the time limit. Since the *sotd* and *ndrr* heuristics are not applicable to most instances with the number of variables more than 5, we remove the comparison with these two heuristics on the results of the new criteria.

There are 1325, 292, 142, 106, 84, 103, and 87 common instances that all heuristics except *sotd* and *ndrr* solved within the time limit for the *random* dataset from 3-var to 9-var; and for the *SMT-LIB* dataset, 1670, 354, 9 instances for 3-var, 4-var to 6-var and 7-var to 9-var, respectively. Table 13 shows the results and the best scores are bolded. We can observe that GRL-SVO still achieves a relatively good performance under the new criteria.

## D.2    Performance of Fine-Tuning

As GRL-SVO(UP) has demonstrated strong generalization abilities and achieved the best performance on almost all datasets, there is still room for further improvement in the generalization capabilities of GRL-SVO(NUP). So, we further investigate the performance of GRL-SVO(NUP) after fine-

Table 12: The performance of different heuristics after the coefficients are randomly modified.

|  | brown | triangular | EMLP | sotd | ndrr | gmods | GRL-SVO(NUP) | GRL-SVO(UP) |
|---|---|---|---|---|---|---|---|---|
| #SI | 831 | 762 | 855 | 898 | 842 | 867 | 892 | **912** |
| AVG.T | 155.42 | 212.50 | 125.54 | 79.73 | 141.29 | 100.30 | 79.96 | **64.89** |
| AVG.N | 2380.67 | 2606.40 | 2353.11 | 2065.13 | 2288.08 | 2162.19 | 2100.57 | **2023.58** |

Table 13: The performance of different heuristics under the new criteria. The dash "-" indicates that the method does not support the category.

| Categories | | NUP | | | | | | UP | |
|---|---|---|---|---|---|---|---|---|---|
| | | EB | | LB | | | | EB | LB |
| | | brown | triangular | EMLP | PVO(brown) | PVO(triangular) | GRL-SVO(NUP) | gmods | GRL-SVO(UP) |
| 3-var(test) | COMAVG.T | 36.45 | 48.77 | 34.08 | - | - | 23.28 | 22.52 | **17.68** |
| | COMAVG.N | 1678.48 | 1968.18 | 1656.93 | - | - | 1474.59 | 1489.72 | **1420.41** |
| 4-var | COMAVG.T | 24.25 | 14.07 | - | 23.22 | 13.75 | 20.50 | **12.48** | 13.82 |
| | COMAVG.N | 2286.79 | 2476.00 | - | 2227.53 | 2306.30 | 2018.79 | **1730.37** | 1739.16 |
| 5-var | COMAVG.T | 27.90 | 32.51 | - | 27.36 | 30.26 | 31.36 | 19.33 | **15.82** |
| | COMAVG.N | 4624.49 | 5222.65 | - | 4355.15 | 4866.49 | 4298.32 | 3415.94 | **3340.59** |
| 6-var | COMAVG.T | 52.71 | 38.95 | - | 56.23 | 40.57 | 38.38 | 32.67 | **22.98** |
| | COMAVG.N | 7687.28 | 6897.06 | - | 7761.30 | 6733.09 | 6281.09 | 5440.62 | **4278.75** |
| 7-var | COMAVG.T | 58.55 | 51.96 | - | - | - | 50.21 | 38.41 | **30.15** |
| | COMAVG.N | 9785.12 | 9055.57 | - | - | - | 8731.50 | 6826.24 | **5810.71** |
| 8-var | COMAVG.T | 102.45 | 118.83 | - | - | - | 91.54 | **67.18** | 69.62 |
| | COMAVG.N | 15484.69 | 17655.02 | - | - | - | 14730.24 | **11053.10** | 12075.62 |
| 9-var | COMAVG.T | 137.26 | 169.25 | - | - | - | 123.67 | 98.77 | **96.90** |
| | COMAVG.N | 20045.44 | 23814.45 | - | - | - | 18132.95 | 16094.36 | **15995.05** |
| SMT-LIB (3-var) | COMAVG.T | 10.41 | 12.07 | 31.72 | - | - | 10.66 | **9.47** | 9.67 |
| | COMAVG.N | 3234.61 | 3746.72 | 6391.53 | - | - | 2906.43 | **2736.89** | 2815.23 |
| SMT-LIB (4-var to 6-var) | COMAVG.T | 39.75 | 41.06 | - | 36.23 | 39.41 | 34.21 | **28.00** | 29.42 |
| | COMAVG.N | 16839.63 | 16815.20 | - | 14730.58 | 15160.26 | 12977.31 | **10847.26** | 12156.08 |
| SMT-LIB (7-var to 9-var) | COMAVG.T | 43.57 | 43.55 | - | - | - | **17.55** | 36.78 | 41.99 |
| | COMAVG.N | 17169.67 | 17169.67 | - | - | - | **6149.89** | 12669.22 | 14919.67 |

Table 14: The performance of **NUP** heuristics, with the performance of GRL-SVO(NUP) after fine-tuning. The dash "-" indicates that the method does not support the category.

| Categories | | NUP | | | | | |
|---|---|---|---|---|---|---|---|
| | | EB | | LB | | | |
| | | brown | triangular | EMLP | PVO(brown) | PVO(triangular) | GRL-SVO(NUP) |
| 3-var(test) | #SI | 1620 | 1504 | 1686 | - | - | **1772** |
| | AVG.T | 170.63 | 227.60 | 140.06 | - | - | **94.07** |
| | AVG.N | 2421.18 | 2663.09 | 2384.29 | - | - | **2157.92** |
| 4-var | #SI | 415 | 376 | - | 408 | 392 | **456** |
| | AVG.T | 352.87 | 394.71 | - | 360.33 | 376.71 | **298.74** |
| | AVG.N | 5258.57 | 5539.73 | - | 5338.45 | 5536.28 | **5060.19** |
| 5-var | #SI | 236 | 202 | - | 242 | 218 | **253** |
| | AVG.T | 435.64 | 495.34 | - | 419.50 | 466.47 | **394.12** |
| | AVG.N | 12465.41 | 13493.00 | - | 11932.31 | 12826.53 | **11586.65** |
| 6-var | #SI | 175 | 149 | - | 180 | 160 | **204** |
| | AVG.T | 500.62 | 551.15 | - | 489.57 | 526.80 | **432.64** |
| | AVG.N | 20487.45 | 20068.22 | - | 20029.06 | **18993.53** | 19407.97 |
| 7-var | #SI | 163 | 118 | - | - | - | **175** |
| | AVG.T | 549.15 | 632.62 | - | - | - | **522.35** |
| | AVG.N | 28552.12 | 29182.78 | - | - | - | **27779.48** |
| 8-var | #SI | 173 | 138 | - | - | - | **177** |
| | AVG.T | 601.17 | 653.60 | - | - | - | **594.77** |
| | AVG.N | 39540.72 | 40902.42 | - | - | - | **38957.71** |
| 9-var | #SI | 151 | 125 | - | - | - | **171** |
| | AVG.T | 651.36 | 691.92 | - | - | - | **619.39** |
| | AVG.N | 48860.36 | 50315.52 | - | - | - | **47406.84** |
| SMT-LIB (3-var) | #SI | **1770** | 1763 | 1675 | - | - | 1768 |
| | AVG.T | **20.33** | 23.68 | 83.09 | - | - | 20.86 |
| | AVG.N | 4449.79 | 5070.46 | 7654.04 | - | - | **4014.59** |
| SMT-LIB (4-var to 6-var) | #SI | **374** | 372 | - | 372 | 372 | 365 |
| | AVG.T | **83.92** | 87.85 | - | 86.22 | 86.88 | 88.42 |
| | AVG.N | 24514.73 | 24178.54 | - | 22953.99 | 22644.04 | **20910.95** |
| SMT-LIB (7-var to 9-var) | #SI | **13** | 12 | - | - | - | 12 |
| | AVG.T | **308.14** | 377.32 | - | - | - | 341.21 |
| | AVG.N | 53971.24 | 58675.94 | - | - | - | **52381.82** |

tuning. We conduct a case study on fine-tuning, utilizing 100 instances with 4 variables. All the hyperparameters employed during the training process were identical to those listed in Table 4.

As shown in Table 14, after fine-tuning, GRL-SVO(NUP) exhibits the best performance among all NUP methods across all categories of the *random* dataset. The $\#SI$ indicator for GRL-SVO(NUP) showed enhancements ranging from 1.00% to 14.38% across the 4-var to 9-var categories of the *random* dataset, with slight improvements also observed in the *SMT-LIB* dataset. Although fine-tuning can help improve the efficiency of GRL-SVO(NUP), the approach is not fit for the instances with a relatively larger number of variables, as we need to run the most variable orders of each instance to build a training dataset. So, for high-dimensional instances, how to fine-tune is a promising direction.

