# OpenReview forum: "Suggesting Variable Order for Cylindrical Algebraic Decomposition via Reinforcement Learning"
_NeurIPS.cc/2023/Conference — NeurIPS 2023 poster_

### Official Review · Reviewer_gZi1 · 2023-07-06

**Soundness:** 3 good
**Presentation:** 3 good
**Contribution:** 3 good
**Rating:** 5
**Confidence:** 3

**Summary:**

Cylindrical algebraic decomposition (CAD) is a technique to decompose a multi-dimensional space into a finite number of cells. CAD cells are built to respect a set of polynomial constraints such that each constraint has constant truth value in each cell. As stated by the authors, the variable ordering in the CAD algorithm significantly influences the computational time, memory usage, and the number of cells. The authors used Reinforcement Learning based approaches to choose an optimal variable ordering. The authors represented a polynomial set in a graph, in addition to the embeddings formed with different indicators of each polynomial (e.g. degree statistics ), and used the number of obtained cells as a quality measure of the result. The authors used the Advantage Actor-Critic (A2C) framework to improve CAD by suggesting a better variable order. The actor neural network was represented by Graph Neural Network as they are permutation invariant.

**Strengths:**

- Reinforcement Learning methods are widely adopted to optimize non-differentiable objectives, and the authors demonstrate its potential in choosing the variable order for CAD algorithms .
- The experimental results demonstrate that their method outperforms the baseline.

**Weaknesses:**

The paper lacks some ablation studies.
- The advantage of using a GNN and a graph representation of polynomial sets: The authors mentioned that GNNs are invariant to permutation, the results should indeed be independent from the order of the polynomial in the set, but there are simpler Neural Network that are invariant to permutation. Can we empirically check how much a GNN enhances the performance of GRL-SVO? To check this, the authors can for instance replace the GNN with an MLP performed on each polynomial embedding and then do a mean aggregation to have a learnable representation of the whole polynomial set.
- I would also encourage authors to test different GNN architectures (e.g. GAT, GATv2, GraphSage ...) for the ablation study.
- What is the effect of normalization factor “M” on the performance and speed of GRL-SVO?
- More experiments should be carried out to assess robustness.
- Code not shared to verify results.

**Questions:**

- What is the used architecture of the GNN? The authors mentioned “GraphConv” and cited the paper [26]. I didn’t see how the two are related to each other. If you are referring to the layer “GraphConv” in the DGL library, the adequate name in the paper is “Garph Convolution Network layer [Kipf and Welling 2017]”.
- Also, the paper [26] is more a method to enhance the expressiveness of a GNN rather than a GNN architecture.

**Limitations:**

- The authors mentioned the limitations of their approaches in section 5.

---

> ### Author Rebuttal · Authors · 2023-08-10
>
> Thank you for the feedback and valuable comments.
>
> ```
> Can we empirically check how much a GNN enhances the performance of GRL-SVO? To check this, the authors can for instance replace the GNN with an MLP performed on each polynomial embedding and then do a mean aggregation to have a learnable representation of the whole polynomial set.
> ```
> GRL-SVO(UP/NUP) accepts the set of variable embeddings, and then GNNs encode the variable embedding into intermediate learnable representations. The actor makes decisions based on this information. We think utilizing polynomial embedding seems to be a new task different from our work. We have conducted an experiment on a simple neural network with MLP layers for nodes without edges. The MLP layer will convert variable embeddings (not polynomial embedding) into their learnable representations. It is also invariant to permutation. It is Experiment 2 (NO_EDGE(MLP) in Table 2 and Figure 1 in PDF attachment) in the response for reviewer HXCT. We will append it to the Appendix.
>
> The main result is that GRL-SVO can outperform these MLP models with different sizes. It is the advantage of the graph structure where a variable can grasp neighbor information.
>
> Combining our work with polynomial embedding for suggesting variable order is also a promising direction, we will investigate it in the future.
>
> ```
> I would also encourage authors to test different GNN architectures (e.g. GAT, GATv2, GraphSage ...) for the ablation study.
> ```
>
> We have arranged experiments on different GNN architectures and will append the results to the Appendix. While the experiments are still ongoing, we have included a subset of the complete experiments on GraphSage in Table 3, which can be found in the PDF attachment.
>
> ```
> What is the effect of normalization factor $M$ on the performance and speed of GRL-SVO?
> ```
> Experiment 6: We make an ablation experiment on $M$ and will append it to the Appendix. We train the models with $M$=10000, 20000, 50000(ours), 100000, and without $M$. Note that if there is no $M$, the reward (the number of cells) will be a relatively large integer. As shown in Table 4, $M$ is necessary. The first number is the result of the validation set, while the second is the result of the testing set.
>
> ```
> Code not shared to verify results.
> ```
> We have submitted our source code to AC. We will clean and release the source code and dataset for the experiments in this paper on GitHub.
>
> ```
> What is the used architecture of the GNN? The authors mentioned ''GraphConv'' and cited the paper [26]. I didn’t see how the two are related to each other. If you are referring to the layer ''GraphConv'' in the DGL library, the adequate name in the paper is ''Graph Convolution Network layer [Kipf and Welling 2017]''.
> Also, the paper [26] is more a method to enhance the expressiveness of a GNN rather than a GNN architecture.
> ```
>
> We apologize for the confusion and will revise it in the next version.
> First, we utilize PyTorch Geometric (PyG) for the implementation in Section 4.1, not the DGL library. There is a difference between the names of the two libraries.
> Second, The operator of GraphConv in PyG is $\mathbf{x}^{\prime} _i = \mathbf{W} _1 \mathbf{x} _i + \mathbf{W} _2 \sum _{j \in \mathcal{N}(i)} e _{j,i} \cdot \mathbf{x} _j$. As there is no edge weight in our models, actually the operator we use is $\mathbf{x}^{\prime} _i = \mathbf{W} _1 \mathbf{x} _i + \mathbf{W} _2 \sum _{j \in \mathcal{N}(i)} \mathbf{x} _j$. Note that a basic GNN model has the (5.7) formula (basic GNN model) in section 5.1.3 of the book ''Graph Representation Learning'' by William L. Hamilton. $$
> \mathbf{h}^{(k)} _u = \sigma(\mathbf{W}^{(k)} _{self} \mathbf{h}^{(k-1)} _u + \mathbf{W}^{(k)} _{neigh}
> \sum _{v \in \mathcal{N}(u)} \mathbf{h}^{(k-1)} _v + b^{(k)})
> $$ Our model is a simple instance of a basic GNN model where $\sigma = relu$, $\mathcal{N}$ means the nodes connecting to $u$, and $\mathbf{W}^{(k)} _{self}$ and $\mathbf{W}^{(k)} _{neigh}$ are all learnable parameters.
> We will revise the citation [26] to (5.7) in ''Graph Representation Learning'' as we used and write down the used GNN formula in the paper.

---

> > ### Author Response · Authors · 2023-08-16
> >
> > With the consent of AC, we provide an anonymous link to the source code: https://anonymous.4open.science/r/GRL-SVO-53C2/.
> >
> > If you have any questions or concerns, we would be delighted to discuss them with you.

---

> > > ### Author Response · Authors · 2023-08-18
> > >
> > > Dear reviewer gZi1,
> > >
> > > We have conducted more experiments by increasing the number of epochs to 100. However, due to the time-consuming nature of interacting with the symbolic computation tool, the experiments on UP have not yet been completed. On the other hand, the experiments on NUP, running for 100 epochs, have been completed. Here, we update the results, including experiments on MLP, other GNN architectures, and analysis on $M$. Note that the performance of GRL-SVO(NUP) in 100 epoches are #SI = 1772, AVG.T = 94.87, AVG.N = 2166.67.
> > >
> > > The following table presents the performance of MLP with various sizes. For instance, ''4-256'' indicates a model with 4 layers and a 256-dimensional intermediate representation. It is important to note that ''4-512'' has twice the number of parameters compared to our model. It shows that the positive effect of graph structure on learning.
> > >
> > > | | 2-256 | 3-256 | 4-256 | 4-512 |
> > > | --- | --- | --- | --- | --- |
> > > | #SI | 1763 | 1763 | 1764 | 1756 |
> > > | AVG.T | 97.89 | 98.43 | 97.68 | 99.08 |
> > > | AVG.N | 2132.33 | 2140.18 | 2129.33 | 2148.64 |
> > >
> > > We test other GNN architectures that accept the same parameters in PyG:
> > > - ClusterGCNConv: operator from "Cluster-GCN: An Efficient Algorithm for Training Deep and Large Graph Convolutional Networks";
> > > - EGConv: operator from "Adaptive Filters and Aggregator Fusion for Efficient Graph Convolutions";
> > > - FiLmConv: operator from "Adaptive Filters and Aggregator Fusion for Efficient Graph Convolutions";
> > > - LEConv: operator modified from "Adaptive Filters and Aggregator Fusion for Efficient Graph Convolutions";
> > > - GATConv: operator from "Graph Attention Networks";
> > > - GATv2Conv: operator from "How Attentive are Graph Attention Networks?";
> > > - GeneralConv: operator from "Design Space for Graph Neural Networks";
> > > - ResGatedGraphConv: operator from "Residual Gated Graph ConvNets";
> > > - SageConv: operator from "Inductive Representation Learning on Large Graphs";
> > > - TransformerConv: operator from "Masked Label Prediction: Unified Message Passing Model for Semi-Supervised Classification".
> > >
> > > | | ClusterGCNConv | EGConv | FiLmConv | LEConv | GATConv | GATv2Conv | GeneralConv |
> > > | --- | --- | --- | --- | --- | --- | --- | --- |
> > > | #SI | 1767 | 1765 | 1760 | 1761 | 1726 | 1762 | 1768 |
> > > | AVG.T | 94.26 | 98.53 | 96.49 | 96.76 | 119.72 | 104.14 | 95.17 |
> > > | AVG.N | 2144.66 | 2177.32 | 2136.67 | 2151.75 | 2166.78 | 2184.86 | 2132.81 |
> > >
> > > ||ResGatedGraphConv | SageConv | TransformerConv |
> > > | --- | --- | --- | --- |
> > > | #SI | 1759 | 1765 | 1769 |
> > > | AVG.T | 96.25 | 96.74 | 94.45 |
> > > | AVG.N | 2146.59 | 2131.22 | 2134.51 |
> > >
> > > We have observed that each graph neural network (GNN) is capable of effectively learning this problem. Our approach is not reliant on the specific network structure.
> > >
> > > From the experiments of $M$, we find that $M$ is necessary. $M$ only affects training process. In the training set, the maximum number of cells for 3-variable instances is 18801. Therefore, selecting $M$ close to or greater than 18801 is recommended.
> > >
> > > When reward is normalized to a relatively small value, it can improve the stability of training and help to converge faster. However, using NO_M or setting $M=1000$ leads to a deteriorating model performance over the training epochs. On the other hand, setting $M=10000, 50000, 100000$ results in a normal training process.
> > >
> > > In actual situations, when $M$ increases, the index of epoch in which the optimal model appears is also delayed. $M=50000$ has the smallest variance of #SI on validation set among these settings.
> > >
> > > | | 10000 | 50000 | 100000 |
> > > | --- | --- | --- | --- |
> > > | index of epoch of optimal model | 37 | 64 | 93 |
> > > | variance on validation set | 94.19 | 24.52 | 30.99 |

---

> > > > ### Comment · Reviewer_gZi1 · 2023-08-22
> > > > **Thank you for the response**
> > > >
> > > > I have read all the comments and analyzed the new experiments conducted by the authors. I am happy to raise my score from 4 to 5.

---

> > > > > ### Author Response · Authors · 2023-08-22
> > > > >
> > > > > Dear Reviewer gZi1,
> > > > >
> > > > > Thank you for taking the time to review our response. We are grateful for your positive feedback.
> > > > >
> > > > > Sincerely,
> > > > >
> > > > > Authors

---

### Official Review · Reviewer_Ydm9 · 2023-07-07

**Soundness:** 3 good
**Presentation:** 3 good
**Contribution:** 2 fair
**Rating:** 6
**Confidence:** 3

**Summary:**

Given a curve y = x^2, it partitions the x-y plane into three sets where the sign y-x^2 is the same. On the curve y-x^2 it is zero, above the curve y = x^2, sign is positive and below the curve it is negative. Thus the polynomial has 3 cells - regions where the sign is invariant.
Given a polynomial with n-variables there is a mathematical procedure (project, root-isolate and lift) to identify cells.  The mathematical procedure may results in different cells depending upon on how the variables are ordered. The objective is to choose a variable ordering in order to minimize the number of cells generated by the mathematical procedure.

The contribution of the paper is to formulate a graph representation of the problem (the state), the action is the variable permutation and the reward is minimizing the number of cells. The  REINFORCE algorithm is used.



**Strengths:**

1. It is nice to see that RL is being used for such problems. It is a nice and natural fit.
2. The model is trained on 3-variable set but generalizes to upto 9 variable set.

**Weaknesses:**

1. The novelty is on setting up the problem. There is not much RL novelty.
2. REINFORCE has high variance. There should be more analysis of the impact of the algorithm on the solution.


**Questions:**

1. The performance improves very quickly according to Figure 4. Is there an explanation. How stable are the runs of REINFORCE ?

---

> ### Author Rebuttal · Authors · 2023-08-10
>
> Thank you for the valuable feedback.
>
> ```
> The performance improves very quickly according to Figure 4. Is there an explanation. How stable are the runs of REINFORCE ?
> ```
>
> As an example, the brown heuristic (EB & NUP in Section 2.2) only utilizes three statistical features: degree, total degree, and occurrence to distinguish the importance of variables, where it can also achieve a good result. GRL-SVO(UP/NUP) focus on a superset of such heuristics. They have a greater probability to construct a relatively effective heuristic by organizing these features, and 3-var instances are relatively small and suitable for learning. Therefore, it is reasonable that the models have a quick improvement in the first few epochs.
>
> Besides, due to the quick improvement caused by the first epoch, the subsequent variations of metrics become very hard to see in Figure 4. We have redrawn the plots starting from the second epoch, referring to Figure 3 in the PDF attachment. It can be observed that REINFORCE exhibits some instability.

---

> ### Comment · Reviewer_Ydm9 · 2023-08-18
>
> I have read all the comments by the authors and other reviewers and am satisfied by the response. I will main my original rating

---

> > ### Author Response · Authors · 2023-08-20
> >
> > Dear Reviewer Ydm9,
> >
> > We appreciate your thorough review and are delighted that our responses have satisfied your queries and concerns.
> >
> > Best regards,
> >
> > Authors

---

### Official Review · Reviewer_zrjH · 2023-07-25

**Soundness:** 2 fair
**Presentation:** 3 good
**Contribution:** 3 good
**Rating:** 7
**Confidence:** 3

**Summary:**

This work proposes a reinforcement learning based method for the selection of a more efficient variable order for the downstream CAD(cylindrical algebraic decomposition) task. The objective is to minimize the number of cells, a suitable metric that intuitively reflects CAD efficiency. The proposed GRL-SVP(UP/NUP) utilizes the inductive biases of GNNs to learn the relationships among the variables in the polynomial set and will output a variable order for CAD via the Actor-Critic algorithm.

Summary:
The main novelty lies in the utilization of RL and GNN for the problem of suggesting variable order for CAD and brings the benefit of better empirical performance and generalization. It contributes to the field as the first work to try RL method for this task and it can be further improved by exploring how to encode polynomial coefficients as edge information, boost prediction time, etc. The inclusion of an interpretive analysis outlining the reasoning behind the variable order proposed by the RL approach would enrich the paper's contribution and reader's understanding.

**Strengths:**

Pros:
1. It effectively reframes the polynomial set as an associated graph, thereby capitalizing on the inherent advantages of GNNs such as permutation invariance and sparse input awareness. This approach may potentially unveil complex variable interrelationships that could otherwise be overlooked by traditional handcrafted heuristics.
2. This study is the first to utilize reinforcement learning to treat this task as a reinforcement problem.
3. Experimental results show compelling evidence for this method when compared to other heuristic methods in the past. And this approach exhibits good generalization when scaled to higher variable problems.
4. The paper seems to be well-structured and provides sufficient detail about the experiments.

**Weaknesses:**

Cons:
1. The associated graph representation may not fully capture all information about the polynomials such as the coefficients. It only encodes relationships between variables. This could be a limitation as the coefficients in a polynomial do carry significant mathematical information.
2. The proposed approach doesn't significantly enhance the inference time compared to existing heuristics as in Figure 4c.

**Questions:**

1. Could you clarify the role of polynomial coefficients within your proposed graph representation? Specifically, are these coefficients incorporated into the graph structure or the node embeddings? Furthermore, what is your perspective on the potential importance of these coefficients for determining optimal variable orderings?
2. Could you elaborate on how GRL-SVO(UP) and GRL-SVO(NUP) are complementary to each other?

**Limitations:**

Yes, the author has adequately addressed limitations in the designated section.

---

> ### Author Rebuttal · Authors · 2023-08-10
>
> Thank you for the valuable feedback.
> ```
> Could you clarify the role of polynomial coefficients within your proposed graph representation? Specifically, are these coefficients incorporated into the graph structure or the node embeddings? Furthermore, what is your perspective on the potential importance of these coefficients for determining optimal variable orderings?
> ```
> Indeed, with regard to the choice of variable order, the current works do not consider coefficients (both Expert-Based or Learning-Based heuristics). Some experts in Symbolic Computation believe that the reason may be related to the calculation of CAD projection: CAD projection uses two polynomials to make a resultant (definition 1 in the Appendix) to eliminate a common variable, so it first depends on the common variable set of these two polynomials (which does not involve coefficients); secondly, the amount of calculation generally depends on the degree of public variables, because the degree determines the size of the resultant matrix.
> For practical instances, the variation range of the coefficient is uncontrollable, which will increase the difficulty of Learning-Based design and training the neural network.
> The impact of coefficients on the problem is complex. We are trying to provide some intuition through Figure 2 in the PDF attachment.
> - (a): $\{ x^3y+4x^2+xy, -x^2+2xy-1 \}, x \prec y: 13, y \prec x: 89$;
> - (b): $\{ x^3y+8x^2+xy, -x^2+2xy-1 \}, x \prec y: 13, y \prec x: 89$;
> - (c): $\{ x^3y-4x^2+xy, -x^2+2xy-1 \}, x \prec y: 45, y \prec x: 125$;
> - (d): $\{ x^3y+4x^2+xy, x^2+2xy-1 \}, x \prec y: 29, y \prec x: 101$;
> - (e): $\{ -x^3y+4x^2+xy, -x^2+2xy-1 \}, x \prec y: 45, y \prec x: 97$.
>
> The number of cells of (a) and (b) are the same, while (c), (d), and (e) are different. But the best variable order is the same ($x \prec y$) in these cases. To a certain extent in these cases, the coefficient mostly affects the number of cells.
>
> Experiment 4: We conduct an experiment on the coefficient and will append it to the Appendix. We have randomly modified the coefficient (in [-100, 100]) of 1000 instances randomly selected from the 3-var testing set. Since the coefficients were the only modification made, we used the original variable order generated from the unaltered instances. Our models continue to outperform other heuristics, as demonstrated by the results obtained.
>
> | | brown | triangular | EMLP | sotd | ndrr | gmods | GRL-SVO(NUP) | GRL-SVO(UP)|
> | --- | --- | --- | --- | --- | --- | --- | --- | --- |
> | #SI | 831 | 762 | 855 | 898 | 842 | 867 | 892 | **912** |
> | AVG.T | 155.42 | 212.50 | 125.54 | 79.73 | 141.29 | 100.30 | 79.96 | **64.89** |
> | AVG.N | 2380.67 | 2606.40 | 2353.11 | 2065.13 | 2288.08 | 2162.19 | 2100.57 | **2023.58** |
>
> We also observe a slight decline in the performance of all heuristics, indicating that the coefficient plays a significant role as a parameter (although it may not be the most crucial one). Effectively analyzing coefficients poses a challenging problem, and we will endeavor to address this issue in the future.
>
> ```
> Could you elaborate on how GRL-SVO(UP) and GRL-SVO(NUP) are complementary to each other?
> ```
> We will revise the unexplained description in the next version. It corresponds to a case where GRL-SVO encounters a large instance. Although GRL-SVO(UP) exhibits superior performance, it is time-consuming due to the involvement of projection and interaction with symbolic calculation tools. As a compromise, we can utilize GRL-SVO(UP) to predict the initial variables for the variable order, while employing GRL-SVO(NUP) for predicting the remaining variables, or vice versa. Alternatively, cross-invoking both methods can also be considered as a viable solution. In this particular case, they complement each other effectively.

---

> > ### Comment · Reviewer_zrjH · 2023-08-18
> > **Thank you for the response**
> >
> > I have read the comments and new experiments conducted by the authors. The fact that their method continuously outperforms other models with varying coefficients shows the method is robust. I am happy to raise my score from 6 to 7.

---

> > > ### Author Response · Authors · 2023-08-20
> > >
> > > Dear Reviewer zrjH,
> > >
> > > We are delighted that our explanations resolve your concerns and appreciate your very encouraging feedback.
> > >
> > > Best regards,
> > >
> > > Authors

---

### Official Review · Reviewer_HXCT · 2023-07-26

**Soundness:** 3 good
**Presentation:** 3 good
**Contribution:** 3 good
**Rating:** 8
**Confidence:** 3

**Summary:**

This paper proposed a new method for suggesting the variables order for Cylindrical Algebraic Decomposition (CAD) problem. Their method utilized Graph Neural Networks (GNN) and Reinforcement Learning(RL). They proposed two variants: one utilizing projection(UP) and one without projection (NUP). They test their methods on two datasets and showed that the UP method outperforms nearly all methods on nearly all tasks, while the NUP method outperforms almost all NUP method. In addition, the UP method they proposed is the first LB and UP method, according to their report.

In summary, I think this is a very creative application of RL and GNN on combinatorial optimization problems, though some improvement and ablation study can be made. The writing is very good, and the comparison to the current methods is complete and well categorized.

**Strengths:**

1. They proposed two algorithm: GRL-SVO(UP) and GRL-SVO(NUP), utilizing the techniques in RL and GNN to optimize the number of cells  of CAD problem for a certain variable order. I think this idea is creative and worth investigating.
2. Their explanation to CAD and SVO(suggesting variable orders) is very clear (especially the detailed version in the appendix), and they gave many example to help people who are not familiar with CAD (like me) to quickly understand what they are doing.
3. The experiments showed that their method outperform most of the current methods on most datasets, and they also compare the UP method and NUP method they developed.
4. The writing is very good and the literature review is clear and complete.

**Weaknesses:**

In general this is a good paper. Below are my personal suggestion for a better paper. I do not expect you to add more experiments during the short rebuttal period, so you should not worry about it.

1. The interpretability is relatively weak. Compared to the Expert based methods, the learning-based method typically use some black-box model such as deep neural networks, GNN or RL to optimize the number of cells of CAD problem. However, more explanation and ablation study can be done, although in general, this GNN+RL method is a black box. For example, in EB methods, people often use some human-created features to suggest the variables order. Since you encode 14 human-crafted features in the graph embedding matrix, maybe you can show which features in the matrix are the most important? Or does this completely depend on the property of the set of polynomials? Also, since you use the adjacent matrix of association graph as well as the variable embeddings as inputs, what will happen if you simply input the adjacent matrix or variable embedding? Will the performance drop drastically?

2. There might be overfitting when you apply a large neural networks on a small set of data. In your experiments, the NN is four-layers with hundreds of hidden size, but the data size is relatively small with only thousands and even hundreds of data. So I am not sure whether there will be overfitting in your training. I think one possible solution is to try to use a smaller neural networks to train the model and evaluate it (also you can try a larger one) to see whether the performance will drop when the size increases. Another question is, I do not see how you use the validation dataset (in Line 269, you said the training : validation : testing is 8:1:1). Typically the validation set is used to tune some hyperparameters, but I do not see in the appendix that you tuned any parameter using this set.

**Questions:**

See above.

**Limitations:**

See above.

---

> ### Author Rebuttal · Authors · 2023-08-10
>
> Thank you for your feedback. Your suggestions are very important and helpful to improve the quality of our work. So we have arranged as many experiments as possible to further discuss our results.
>
> ```
> Since you encode 14 human-crafted features in the graph embedding matrix, maybe you can show which features in the matrix are the most important?
> ```
> Experiment 1: We conduct an experiment on the effect of features and will append it to the Appendix. We make masks for these 14 features, where the mask will set the features that we do not care about as zero:
> 1. One-hot masks (test the effect of a single feature), for example, to test the effect of $E_1$, the corresponding one-hot mask is (1, 0, 0, 0, 0, 0, 0, 0, 0, 0, 0, 0, 0, 0). Multiplying with input feature will result in a feature vector with only $E_1$ while others are 0.
> 2. Operation masks (test the effect of different operations in features) will group features according to their operation type (maximum, sum, and proportion). Note that we treat $E_1, E_2$ as sub-features utilizing sum operation.
> -  Max: $E_3,E_5,E_6,E_7,E_{13},E_{14}$
> -  Sum: $E_1,E_2,E_4,E_8,E_9,E_{10}$
> -  Prop: $E_{11},E_{12}$
> 3. Object masks (test the effect of different objects in features) will group features according to their target objects (variable, term, and polynomials).
> -  Var: $E_1,E_3,E_4,E_{13},E_{14}$
> -  Term: $E_5,E_6,E_7,E_8,E_9,E_{10},E_{12}$
> -  Poly: $E_2,E_{11}$
> 4. Because degree is a common feature that most heuristics concern, testing the effect of degree is necessary. Degree masks will group features according to whether utilize degree.
> -  Degree: $E_3,E_4,E_5,E_7,E_8,E_9$
> -  NoDegree: $E_1,E_2,E_6,E_{10},E_{11},E_{12},E_{13},E_{14}$
>
> Due to space limitations, we do not list the results of a single feature. It shows that only one feature is not enough. The best performing feature is $E_{12}$ (on GRL-SVO(UP)) with #SI = 1706, AVG.T = 129.86, AVG.N = 2386.24. Table 1 in the PDF attachment shows the results of operation, object, and degree features.
>
> The sum, term, and degree may be the most important factors, as whether using Sum/Term/Degree features will result in the largest difference on performance.
>
> ```
> since you use the adjacent matrix of association graph as well as the variable embeddings as inputs, what will happen if you simply input the adjacent matrix or variable embedding? Will the performance drop drastically?
> ```
> Experiment 2: We conduct an experiment on the effect of architecture and will append it to the Appendix. We build two models: one without embedding (NO_EMB), and the other without edge (NO_EDGE). Note that the operator of GraphConv in PyG is $\mathbf{x}^{\prime} _i = \mathbf{W} _1 \mathbf{x} _i + \mathbf{W} _2 \sum _{j \in \mathcal{N}(i)} e _{j,i} \cdot \mathbf{x} _j$. As there is no edge weight in our models, actually the operator we use is $\mathbf{x}^{\prime}_i = \mathbf{W} _1 \mathbf{x} _i + \mathbf{W} _2 \sum _{j \in \mathcal{N}(i)} \mathbf{x} _j$. NO_EDGE will lead $\mathcal{N}$ to be empty, and $W_2$ will lose its effect, so it will be MLPs. Table 2 in the PDF attachment shows the results of NO_EDGE(MLP) and NO_EMB.
>
> The performance of NO_EMB drops dramatically while that of NO_EDGE is good. GRL-SVO can still outperform such models and GRL-SVO(NUP) outperforms much more. Due to NUP training time is relatively short (compared to UP), we continue to train GRL-SVO(NUP) and NO_EDGE(NUP) to 30 epochs and explore the performance of NO_EDGE under different sizes of parameters. MLP_4_512 has twice as many parameters as ours. GRL-SVO(NUP) can outperform all MLPs as shown in Figure 1. It is the advantage of the graph structure where a variable can grasp neighbor information.
>
>
> ```
> I think one possible solution is to try to use a smaller neural networks to train the model and evaluate it (also you can try a larger one) to see whether the performance will drop when the size increases.
> ```
> Experiment 3: We have conducted an experiment on the effect of size and will append it to the Appendix. # GNN layers (for short, #G) $\in$ {1, 2, 3, 4, 5} and # Intermidiate layer features (for short, #I) $\in$ {32, 64, 128, 256, 512}$ where Actor and Critic will keep the same proportion 2:4:1. Note that the input dimension of Actor and Critic is the same as #I. For example, if #I = 32, then the dimension of the Actor and Critic are both [32, 64, 16]. Elements in the following tables are #SI in the validation set, #SI in the testing set, respectively. The first table is the performance (#SI) of GRL-SVO(NUP) and the second is the performance (#SI) of GRL-SVO(UP).
>
>
> | | 32 | 64 | 128 | 256 | 512 |
> | --- | --- | --- | --- | --- | --- |
> | 1 | 1754,1737 | 1756,1741 | **1771**,1765 | 1770,1765 | 1764,1765 |
> | 2 | 1771,1764 | 1762,1747 | 1765,1763 | 1767,1766 | 1764,1768 |
> | 3 | 1765,1758 | 1756,1744 | 1766,1758 | 1761,1766 | 1770,**1769** |
> | 4 | 1765,1761 | 1768,1761 | 1764,1762 | 1766,1765 | 1769,1765 |
> | 5 | 1768,1762 | 1765,1763 | 1771,1759 | 1766,1761 | 1763,1762 |
>
>
> | | 32 | 64 | 128 | 256 | 512 |
> | --- | --- | --- | --- | --- | --- |
> | 1 | 1767,1759 | 1768,1758 | 1777,1774 | 1789,1789 | 1790,1793 |
> | 2 | 1773,1763 | 1780,1777 | 1778,1776 | 1786,1793 | 1788,**1794** |
> | 3 | 1781,1767 | 1783,1778 | 1782,1785 | 1788,**1794** | 1786,1793 |
> | 4 | 1775,1766 | 1784,1775 | 1780,1779 | **1792**,**1794** | 1788,1793 |
> | 5 | 1779,1779 | 1784,1781 | 1788,1797 | **1792**,1790 | 1791,1790 |
>
> The results show that (4,256) seems a better option. If the size is larger than (4, 256), i.e., (4, 512), (5, 256), (5, 512), the effect of the network dropped slightly.
>
> ```
> I do not see how you use the validation dataset (in Line 269, you said the training : validation : testing is 8:1:1).
> ```
> We have selected the best model parameters during training via the validation dataset and will append the results of selecting hyperparameters to the Appendix.

---

> > ### Comment · Reviewer_HXCT · 2023-08-22
> >
> > Thanks the authors for their additional experiments! I think these ablation study are very good so raised my score to eight.
> >
> > I will strongly support this paper to be accepted, given their creative and solid work and the fact that they added many experiments in the short rebuttal period of time.
> >
> > Good luck!

---

> > > ### Author Response · Authors · 2023-08-22
> > >
> > > Dear Reviewer HXCT,
> > >
> > > We are delighted that our responses have met your satisfaction. We would like to express our heartfelt gratitude for your recognition and support regarding our submission.
> > >
> > > Sincerely,
> > >
> > > Authors

---

### Author Rebuttal · Authors · 2023-08-10

We would like to express our sincere gratitude for the reviewers' valuable feedbacks, which are quite helpful to enhance the quality and clarity of our work. We have designed some experiments and attempted to complete them as many as possible during this short period. We will add the missing experiments that the reviewers are concerned about to the Appendix. The PDF attachment contains some tables and figures.

**Notes on Additional Experiments**
If not specified otherwise, all models are trained on the 3-var set in 10 epochs. The best model parameters are selected on the 3-var validation set, and the models are tested on the 3-var testing set. GRL-SVO (UP/NUP) are the models trained in 10 epochs for fair comparison. The performance of GRL-SVO(NUP) and GRL-SVO(UP) for 3-var testing set in 10 epoches are #SI = 1765, AVG.T = 97.78, AVG.N = 2142.98; #SI = 1794, AVG.T = 79.57, AVG.N = 2075.23, respectively.

---

### Decision · Program_Chairs · 2023-09-21

**Decision:**

Accept (poster)

**Comment:**

The reviewers agreed the proposed two algorithms, GRL-SVO(UP) and GRL-SVO(NUP), utilizing the techniques in RL and GNN to optimize the number of cells of CAD problem for a certain variable order, are creative and worth investigating. The empirical results are also solid. I recommend the paper be accepted. Please incorporate the reviewers' suggestions as promised.